# GTool: Graph Enhanced Tool Planning with Large Language Model

**Wenjie Chen**[1,2], **Di Yao**[1,2,*], **Wenbin Li**[1,2], **Xuying Meng**[1], **Chang Gong**[1,2], **Jingping Bi**[1,2]

[1]State Key Laboratory of AI Safety, Institute of Computing Technology,
Chinese Academy of Sciences, Beijing, China
[2]University of Chinese Academy of Sciences, Beijing, China
{chenwenjie23s,yaodi,liwenbin20z,mengxuying,gongchang21z,bjp}@ict.ac.cn

## ABSTRACT

Tool planning with large language models (LLMs), referring to selecting, organizing, and preparing the tools necessary to complete a user request, bridges the gap between natural language understanding and task execution. However, current works treat different tools as isolated components and fail to leverage the inherent dependencies of tools, leading to invalid planning results. Since tool dependencies are often incomplete, it becomes challenging for LLMs to accurately identify the appropriate tools required by a user request, especially when confronted with a large toolset. To solve this challenge, we propose GTool, which is the first work aiming to enhance the tool planning ability of LLMs under incomplete dependencies. GTool constructs a request-specific tool graph to select tools efficiently and generate the `<graph token>` which provides sufficient dependency information understandable by LLMs. Moreover, a missing dependency prediction task is designed to improve the reliability of GTool with incomplete dependencies. Without trimming LLMs, GTool can be seamlessly integrated with various LLM backbones without extensive retraining. Extensive experiments show that GTool achieves more than 29.6% performance improvements compared with the state-of-the-art (SOTA) baselines with a light-weight (7B) LLM backbone.

## 1 INTRODUCTION

Current large language models (LLMs) have achieved significant breakthroughs in a range of natural language processing tasks, but often struggle with numeric computations and delivering accurate, timely information for solving complex problems. Tool planning Qu et al. (2025), enabling LLMs to dynamically interact with external tools, such as APIs and algorithms, is the fundamental ability to improve the problem-solving capability of LLMs Qin et al. (2024a). However, tools are not independent of each other. The input of one tool may depend on the results of other tools, forming the complex tool dependencies. Generating a dependency-correct plan would not only improve the reliability of LLMs, but also shed light on many applications, from general AI systems to industrial applications Huang et al. (2024a).

To capture tool dependencies, existing works can be categorized into two groups: tuning-free methods and tuning-based methods. Tuning-free approaches Paranjape et al. (2023); Schick et al. (2023); Shen et al. (2023); Liu et al. (2024b); Song et al. (2023) focus on prompt design to encode useful tool information as the context of LLM input. Various techniques, *e.g.*, few-shot learning Paranjape et al. (2023), coarse-to-fine strategy Song et al. (2023) and searching on decision tree Zhuang et al. (2023) are designed to combine the user requests, tool descriptions and dependencies for improving the tool planning ability. Without any parameter optimization, tuning-free methods may fail in understanding the user intentions and the context lengths are usually too long to be captured, leading to suboptimal planning performance. On the other hand, tuning-based methods Lumer et al. (2025); Yin et al. (2025); Zhang et al. (2025) either introduce new trainable modules or construct specialized corpora to fine-tune existing LLMs. LLMs are fine-tuned with LoRA Yang et al. (2024b),

---

* Corresponding author.

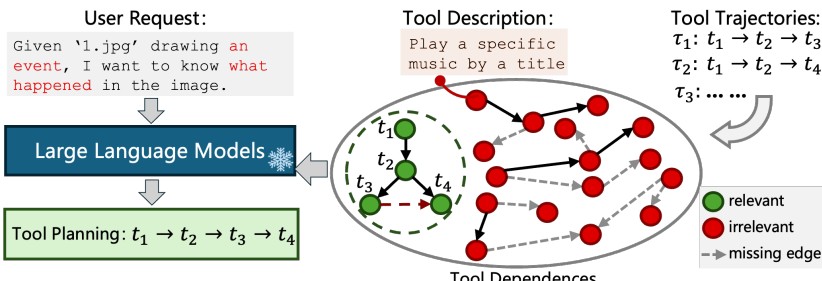

Figure 1: The motivation and challenges of GTool.

Reinforcement Learning from Human Feedback (RLHF) Liang et al. (2024), to achieve better planning performance. Nevertheless, these methods rely on predefined dependency structures, which are often impractical to obtain in real-world scenarios. Moreover, high computational resources are required by these methods and the construction of tuning corpus is labor-intensive.

Thus, tool planning is still in the experimental stage and not yet ready to fully meet real-world demands. Both tuning-free and tuning-based approaches ignore the incompleteness of tool dependencies, resulting in invalid and suboptimal plans. The tool dependencies are usually collected by the invoking tool trajectories. As shown in Figure 1, according to $\tau_1$ and $\tau_2$, we can observe that $t_3$ and $t_4$ must be executed subsequent to $t_2$. The dependency between $t_3$ and $t_4$ is missing. It is intractable to collect sufficient tool trajectory to cover all tool dependencies. In practice, tool dependencies can be naturally represented as a graph, where each node corresponds to a distinct tool, and each edge indicates a valid dependency between tools. Inspired by this observation, we believe that the tool dependency graph is critical for tool planning and suitable for modeling the missing tool dependencies, as massive existing graph learning worksKipf & Welling (2016); Zhang & Chen (2018); Chami et al. (2019) can be used for handling the incomplete dependencies.

However, integrating the tool dependency graph into tool planning is not trivial. It presents the following two challenges: (1) **Request-specified planning.** A notable characteristic of tool dependencies is that they are request-specific, *i.e.*, the related tools vary significantly depending on the tasks described in the user request. How to construct build a request-specified tool graph remains a challenge and would be more apparent in large-scale tool sets. (2) **Modality gap of tool graph.** LLMs are designed to take textual input. Effectively aligning the tool dependency graph and ensuring that its dependencies are properly understood by LLMs remain critical challenges.

To overcome these challenges, we introduce GTool which is the first work specifically designed to model the incomplete tool dependencies and enhance the tool planning performance of LLMs. For request-specified planning, GTool generates a request-specific tool graph which involves the user-request as a request-specific node connecting with all existing tools. The tool descriptions and request context are treated as the features of the dependency graph. Subsequently, a GNN-based module is designed to obtain the <graph token> which provides almost all dependency information. GTool introduces a missing dependency prediction strategy using LLMs, which does not assume that the tool graphs are complete. Additionally, GTool aligns the semantic space of graph representations with tool planning through supervised instruction tuning, while leveraging graph-based dependency to be understood by LLMs.

We conduct comprehensive experiments on four public datasets and summarize the key findings as follows: (1) **Accurate.** GTool significantly outperforms the state-of-the-art tool planning methods, achieving over 29.6% performance improvements with a light-weight (7B) LLM backbone. (2) **Robust.** GTool is robust to missing tool dependencies and can effectively handle sparse tool graphs. Even the missing 90% tool dependencies, GTool can also achieve remarkable performance. (3) **Efficient.** GTool integrates tool descriptions into the <grpah token>, which reduces over 95% pre-task tokens processed by LLMs compared with methods encoding them in prompts. During inference, the time cost of GTool is only one-tenth of SOTA baselines. (4) **Generalizable.** We freeze all parameters of LLMs and only train GNN encoder. Thus, GTool can be seamlessly integrated with various LLMs backbones without extensive retraining.

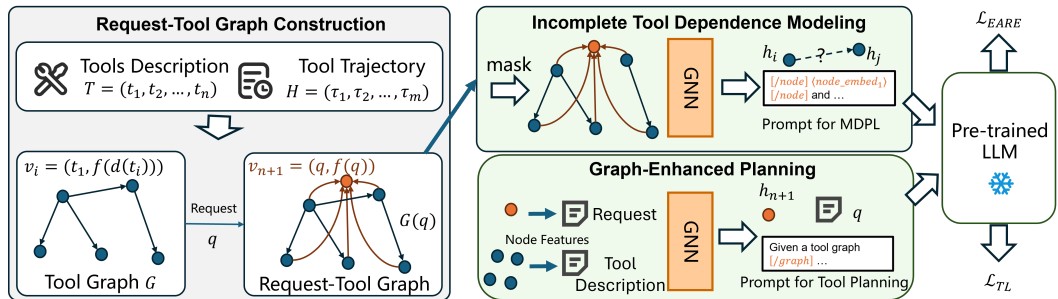

Figure 2: Overview of GTool.

## 2 PROBLEM FORMULATION

**DEFINITION 1** (Tool): *A tool $t$ is defined as an entity,* e.g., *an API, designed to achieve a specific functionality. Each tool $t$ is accompanied by a document $d(t)$, which provides a detailed and formal description of its capabilities, input-output specifications, and operational constraints.*

**DEFINITION 2** (Tool Trajectory): *A tool trajectory refers to an ordered sequence of tools invoked by the an agent or user to fulfill a user request. Formally, a tool trajectory $\tau$ can be represented as $\tau = \{t_1, \ldots, t_{|\tau|}\}$, where each $t_i$ denotes a tool invoked at step $i$.*

**DEFINITION 3** (Tool Graph): *Given a tool set $T$, a tool graph $G = \{V, E\}$ is constructed to represent the dependencies among these tools. In this graph, each node $v_i \in V$ corresponds to a tool $t_i \in T$, and a directed edge $(v_i, v_j) \in E$ indicates a dependency relationship between tools $t_i$ and $t_j$. Specifically, the presence of an edge $(v_i, v_j)$ signifies that the functionality or output of tool $t_i$ is required as an input or precondition for the execution of tool $t_j$.*

**DEFINITION 4** (Graph-Enhanced Tool Planning): *Given a collection of tools $T = \{t_1, \ldots, t_n\}$, a set of historical user requests and corresponding tool trajectories $Q = \{q_1, \ldots, q_m\}$ and $H = \{\tau_1, \ldots, \tau_m\}$, graph-enhanced tool planning first infers interdependencies among tools and construct a tool graph $G$. Subsequently, for each new user request $q'$, graph-enhanced tool planning generates a sequence of tool invocations $\tau'$ leveraging the tool graph $G$.*

## 3 METHODOLOGY

As illustrated in the overview in Figure 2, GTool consists of three modules, *i.e.*, request-specified tool graph construction, tool dependency modeling, and graph-enhanced planning.

### 3.1 REQUEST-SPECIFIED TOOL GRAPH CONSTRUCTION

#### 3.1.1 TOOL GRAPH

The tool graph captures inter-dependencies between tools, enabling LLM to comprehend the complex interactions among a large number of tools in real-world applications, thereby facilitating accurate tool planning. Whereas such a tool graph is often unavailable in practice, we propose to construct a tool graph $G = \{V, E\}$ for a given set of tools $T = \{t_1, \ldots, t_n\}$ based on historical tool trajectories $H = \{\tau_1, \ldots, \tau_m\}$. Formally, the node set $V$ is defined as: $V = \{v_i = (t_i, \boldsymbol{a}_i)|i = 1, \ldots, n\}$, where $t_i$ denotes the associated tool and $\boldsymbol{a}_i$ represents the node's attributes.

To initialize the attribute $\boldsymbol{a}_i$, we input the document $d(t_i)$ of tool $t_i$ into a language model $f$, *e.g.*, BERT, and utilize the generated embedding as the value of $\boldsymbol{a}_i$, *i.e.*, $\boldsymbol{a}_i = f(d(t_i))$. This approach leverages the semantic richness of the tool descriptions to encode meaningful attributes into the graph structure, enhancing the model's ability to capture functional and contextual relationships among tools.

The edge set $E$ is constructed through an iterative analysis of historical tool trajectories. Specifically, we initialize the edge set $E$ as an empty set, and for each trajectory $\tau_i = \{t_{i_1}, t_{i_2}, \ldots\}$, we add following edges to the edge set $E$:

$$E \leftarrow E \cup \{(v_{i_j}, v_{i_{j+1}})|1 \leq j \leq |\tau_i| - 1\}.$$

The orders of tools in historical trajectories does not fully represent the dependency relationships among them. Therefore, the constructed graph is bound to contain both erroneous and missing edges, which is is an unavoidable compromise in the absence of groundtruth. Nevertheless, our method remains applicable in scenarios where groundtruth is available.

### 3.1.2 REQUEST-SPECIFIED TOOL GRAPH.

The constructed tool graph $G$ encodes both the semantic representations of tools and the inter-dependencies among them, providing valuable structural context for tool planning. However, the number of tools required for a given request is significantly smaller than the overall size of the tool graph. Indiscriminately modeling the entire tool graph directly may introduce substantial irrelevant information from unrelated nodes and fails to capture the task-specific nature of tool dependencies, which can hinder the agent's decision-making accuracy.

To address this limitation, for each user request $q$, we generate a request-tool graph $G(q) = \{V(q), E(q)\}$. Specifically, we augment the node set $V(q)$ by introducing a request-specific node $v_{n+1}$ to represent the user request $q$. The text of $q$ is then processed through the language model $f$ and the output is assigned as the node attribute for $v_{n+1}$: $V(q) = V \cup \{v_{n+1} = (q, \boldsymbol{a}_{n+1})|\boldsymbol{a}_{n+1} = f(q)\}$.

Subsequently, we connect directed edges from all other nodes $V(q)$ to the request-specific node $v_{n+1}$: $E(q) = E \cup \{(v_i, v_{n+1})|v_i \in V(q), v_i \neq v_{n+1}\}$. This structural configuration facilitates the propagation of tool semantic information and dependency relationships critical to the user request $q$ along these edges to $v_{n+1}$ during subsequent modeling. As a result, the graph enables the model to aggregate and consolidate request-specific key information, thus enhancing tool planning.

## 3.2 TOOL DEPENDENCE MODELING

### 3.2.1 GRAPH ENCODING

For a given user request $q$, we construct a request-tool graph $G(q)$, and employ a GNN-based encoder with parameters $\theta$ to model both the structural and semantic information embedded within the graph:

$$[\boldsymbol{h}_1, \ldots, \boldsymbol{h}_{n+1}] = \phi(G(q); \theta), \tag{1}$$

where $\boldsymbol{h}_i$ represents the learned representation of node $v_i$ derived by the model.

As every node $v_i, 1 \leq i \leq n$ in the graph $G(q)$ is connected to $v_{n+1}$ via directed edges, critical information relevant to the user request $q$ can propagate along these edges to $v_{n+1}$. Consequently, we designate the representation $\boldsymbol{h}_{n+1}$ of the request-specific node $v_{n+1}$ as the graph representation $\boldsymbol{h}_G$ in the context of user request $q$, i.e. $\boldsymbol{h}_G = \boldsymbol{h}_{n+1}$.

### 3.2.2 MISSING DEPENDENCY PREDICTION

The quality of the constructed graph significantly impacts the performance of tool planning. However, the inter-dependencies among tools may be only partially captured from historical tool trajectories, potentially resulting in the tool graph $G$ and its request-specific variant $G(q)$ being incomplete. To address this issue, we propose missing dependency prediction with LLMs (MDPL) to improve the LLM-based agent's robustness to missing edges in the graph. By explicitly modeling the edge incompleteness, MDPL ensures robust tool planning with partially observed or incomplete dependencies.

Specifically, the missing dependency prediction consists of the following three steps. Firstly, for any pair of nodes $v_i, v_j \in V, v_i \neq v_j$, if there is an edge $i.e.$, $(v_i, v_j) \in E$, we mask it with the probability of $\rho$, simultaneously masking the corresponding edge $(v_i, v_j) \in E(q)$, and add $(v_i, v_j, l = \text{'yes'})$ to the set of positive candidate edges $\hat{E}^+$. Otherwise, if $(v_i, v_j) \notin E$, we add $(v_i, v_j, l = \text{'no'})$

| [/node] ⟨node_embed₁⟩ [/node] and [/node] ⟨node_embed₂⟩ [/node] are two node vectors encoded by the graph neural network, and determines whether the two nodes have edge connections, and only answers yes or no. | Given a tool graph [/graph] ⟨graph_embed⟩ [/graph] and a list of tools ⟨tool_list⟩. ⟨user_request⟩. Please use the provided tools to solve the problem. Just answer the tool in the order they are provided. |
|---|---|
| (a) | (b) |

Figure 3: The prompt instructions of GTool. (a) missing dependency prediction; (2): graph-enhanced tool planning.

to the set of negative candidate edges $\hat{E}^-$. Secondly, for $(v_i, v_j, l) \in (\hat{E}^+ \cup \hat{E}^-)$, we generate a text corpus $x$ following the prompt template illustrated in Figure 3(a), where $\langle node\_embed_1 \rangle = v_i$ , $\langle node\_embed_2 \rangle = v_j$. The token $[/node]$ is used as a special marker to indicate representation boundaries during LLM-based text generation. Thirdly, we train the model with predicting the existence of edges in $\hat{E}$ based on the learned node representations. Moreover, the text corpus $x$ is fed into a large language model $M$ to perform autoregressive supervised fine-tuning, thereby optimizing the parameters $\theta$ of GNN encoder.

This introduces a critical scalability challenge: tool graphs with substantial edge cardinality, direct computation of autoregressive loss over all edges incurs prohibitive computational overhead. To resolve this bottleneck, we sample from positive and negative edge sets in a balanced way:

$$S = \mathrm{RS}(\hat{E}^+, \alpha) \cup \mathrm{RS}(\hat{E}^-, \alpha). \tag{2}$$

where RS denotes the sampling function and hypermeter $\alpha$ denotes the sampling size. For $(v_i, v_j, l) \in S$ the loss function is computed exclusively over the sampled edge set $S$:

$$\mathcal{L}_{MDPL} = \frac{1}{|S|} \sum_S p_M(l|x). \tag{3}$$

where $p_M$ represents the token prediction loss computed by the language model $M$ over the input and label.

### 3.3 Graph-Enhanced Planning

After modeling the request-specific dependencies among tools through graph learning, we employ a large language model to perform tool planning for the user request $q$. We construct a prompt $w$ for the large language model $M$ by integrating the user request $q$, the available tools $T$, the ground truth $\tau_G$, and the representations of $G(q)$ $\boldsymbol{h}_G$. Figure 3(b) illustrates the design of the prompt template employed in GTool, where $\langle tool\_list \rangle$ denotes the list encompassing the names of all tools in $T$, $\langle user\_query \rangle$ denotes the request $q$, $[/graph]$ serves as a special token identifier for the graph representation within the LLM generating. Notably, we assign the $\langle graph\_embed \rangle$ with $\boldsymbol{h}_G$. This strategic implementation enables the large model's reasoning process to effectively integrate both the request and the graph structure, thereby minimizing the influence of irrelevant tools on the performance. We input $w$ into the large language model $M$ and optimize $\theta$ via the following loss:

$$\mathcal{L}_{TL} = p_M(\tau_G|w). \tag{4}$$

The two kinds of losses are summarized with a hyperparameter $\lambda$ to balance the loss scales. Based on Eq. 3 and Eq. 4, we added them to obtain the final loss: $\mathcal{L} = \mathcal{L}_{TL} + \lambda \mathcal{L}_{MDPL}$. This joint training approach allows GTool to effectively learn both tool dependency by tool dependency prediction via $\mathcal{L}_{MDPL}$ and tool planning via $\mathcal{L}_{TL}$, enhancing its overall performance. For a new request $q'$, we perform graph-enhanced tool planning to generate a tool trajectory $\tau'$ for it. Specifically, we input the request $q$ into the language model $f$ for its embedding and construct the request-tool graph $G(q')$ as stated in Section 3.1.2. Then, we generate the graph embedding $\boldsymbol{h}'_G$ to summarize request-specific tool inter-dependencies via the GNN-based encoder as shown in Eq. 1. After that, the graph embedding $\boldsymbol{h}'_G$, list of tool names, and user request $q$ are all input into the LLM $M$ in the format depicted in Figure 3(b), enabling the model to perform tool planning based on the tool dependencies. Finally, we extract the tool trajectory $\tau'$ from the autoregressively generated response of the large language model.

## 4 Experimental Setup

**Datasets.** We conducted comprehensive experiments on two publicly available datasets, *i.e.*, TaskBench Slaughter et al. (2020) and ToolE Huang et al. (2023). TaskBench comprises three

| LLMs | Methods | HuggingFace | | | Daily Life | | | Multimedia | | | ToolE | | |
|---|---|---|---|---|---|---|---|---|---|---|---|---|---|
| | | n-F1 ↑ | l-F1 ↑ | NED ↓ | n-F1 ↑ | l-F1 ↑ | NED ↓ | n-F1 ↑ | l-F1 ↑ | NED ↓ | n-F1 ↑ | l-F1 ↑ | NED ↓ |
| NA | BM25 | 0.4310 | 0.0442 | 0.6654 | 0.5186 | 0.0679 | 0.6162 | 0.3625 | 0.0283 | 0.6983 | 0.2771 | 0.0169 | 0.7580 |
| | COLT | 0.3292 | 0.0263 | 0.7341 | 0.3404 | 0.0255 | 0.7015 | 0.3321 | 0.0233 | 0.7247 | 0.5605 | 0.1481 | 0.5568 |
| | GRTF | 0.3526 | 0.0295 | 0.7193 | 0.2460 | 0.0021 | 0.7922 | 0.2675 | 0.0236 | 0.7543 | 0.4004 | 0.0221 | 0.6519 |
| Llama | TaskBench | 0.4095 | 0.1584 | 0.6033 | 0.2529 | 0.1359 | 0.7605 | 0.3130 | 0.0860 | 0.6997 | 0.0808 | 0.0063 | 0.9344 |
| | HuggingGPT | 0.4457 | 0.1192 | 0.5897 | 0.6229 | 0.3103 | 0.3930 | 0.3399 | 0.0519 | 0.6893 | 0.6184 | 0.2180 | 0.4336 |
| | GNN4Plan | 0.4853 | 0.2418 | 0.5267 | 0.3588 | 0.1940 | 0.6502 | 0.4593 | 0.2311 | 0.5534 | 0.5069 | 0.1870 | 0.5480 |
| | ToolNet | 0.2140 | 0.0096 | 0.8062 | 0.1196 | 0.0020 | 0.8869 | 0.1454 | 0.0057 | 0.8593 | 0.2597 | 0.0139 | 0.7434 |
| | Tool-Planner | 0.2690 | 0.0315 | 0.7726 | 0.2027 | 0.0190 | 0.8153 | 0.2307 | 0.0301 | 0.8040 | 0.3501 | 0.0448 | 0.7638 |
| | GTool | **0.7913** | **0.5403** | **0.2537** | **0.9458** | **0.8375** | **0.0756** | **0.8001** | **0.5855** | **0.2254** | **0.8017** | **0.3800** | **0.3167** |
| Vicuna | TaskBench | 0.4954 | 0.2181 | 0.5226 | 0.7069 | 0.4800 | 0.3176 | 0.2411 | 0.1069 | 0.7655 | 0.7044 | 0.3038 | 0.3769 |
| | HuggingGPT | 0.5079 | 0.1965 | 0.5127 | 0.7449 | 0.5342 | 0.2725 | 0.5111 | 0.1994 | 0.5127 | 0.7500 | 0.3740 | 0.3374 |
| | GNN4Plan | 0.5776 | 0.2978 | 0.4378 | 0.7872 | 0.5637 | 0.2386 | 0.6364 | 0.4021 | 0.3777 | 0.7209 | 0.3132 | 0.3650 |
| | ToolNet | 0.3441 | 0.0423 | 0.7322 | 0.3412 | 0.0330 | 0.7237 | 0.3273 | 0.0568 | 0.7143 | 0.4415 | 0.0423 | 0.6986 |
| | Tool-Planner | 0.3990 | 0.0830 | 0.6622 | 0.3139 | 0.0584 | 0.7108 | 0.2756 | 0.0370 | 0.7542 | 0.4440 | 0.1071 | 0.6427 |
| | GTool | **0.8029** | **0.5816** | **0.2153** | **0.9612** | **0.8638** | **0.0581** | **0.7905** | **0.5694** | 0.2363 | **0.7833** | 0.3500 | 0.3392 |
| Qwen3 | TaskBench | 0.7682 | 0.5645 | 0.2627 | 0.9414 | 0.8311 | 0.0855 | 0.7079 | 0.5586 | 0.3193 | 0.7144 | **0.4227** | 0.3632 |
| | HuggingGPT | 0.7408 | 0.5126 | 0.2727 | 0.9252 | 0.8272 | 0.0944 | 0.7089 | 0.5421 | 0.3055 | 0.7656 | 0.3776 | **0.2946** |
| | GNN4Plan | 0.7602 | 0.5347 | 0.2481 | 0.9024 | 0.7428 | 0.1207 | 0.8269 | 0.6563 | 0.1854 | 0.7639 | 0.4089 | 0.3186 |
| | ToolNet | 0.3543 | 0.0916 | 0.6837 | 0.3197 | 0.0220 | 0.6938 | 0.3154 | 0.0765 | 0.7190 | 0.4111 | 0.0274 | 0.7282 |
| | Tool-Planner | 0.6150 | 0.2755 | 0.4280 | 0.3229 | 0.0255 | 0.6808 | 0.4222 | 0.0989 | 0.5834 | 0.5957 | 0.1258 | 0.4809 |
| | GTool | **0.8053** | **0.5905** | **0.2136** | **0.9668** | **0.8837** | **0.0521** | **0.8543** | **0.6749** | **0.1642** | **0.7749** | 0.4090 | 0.3013 |

Table 1: The planning performance of GTool and baselines on benchmark datasets. ”GRTF” refers to Graph RAG-Tool Fusion.

distinct datasets, where HuggingFace encompasses a collection of AI models serving as tools, Daily Life focuses on real-life scenarios and Multimedia includes user-centric multimedia tools. ToolE covers a diverse range of tools spanning multiple domains and request types. The detailed statistics of the datasets are summarized in the Appendix A.1.

**Baselines.** Three categories of baselines, *i.e.*, naïve methods, tuning-free methods and tuning-based methods, are employed in our experiments. Without using LLMs, naïve methods including **BM25** Robertson et al. (2009). **COLT** Qu et al. (2024) retrieves the most similar tools of requests based on the tool descriptions. **Graph RAG-Tool Fusion** Lumer et al. (2025) takes into account inter-tool dependencies by planning the tool invocation sequence via a depth-first traversal over the dependency graph. Tuning-free methods, such as **HuggingGPT** Shen et al. (2023) and **TaskBench** Slaughter et al. (2020), integrate tool descriptions into prompts and carry out plannings without any parameter optimization. For tuning-based methods, we choose the latest **GNN4Plan** Wu et al. (2024), **ToolNet** Liu et al. (2024a) and **Tool-Planner** Liu et al. (2025) that learn an alignment module for boosting the planning performance of LLMs. Note that GTool only interact with LLMs once for one request. We do not compare works that require multiple LLM interactions, such as STE Wang et al. (2024b), Toolink Qian et al. (2023a) and ToolLLaMA Qin et al. (2024b). These works may further improve the performance of GTool and we leave them as future work. More details of the compared baselines are in Appendix A.2.

**LLM Backbones.** We evaluate the performance of GTool on ten open-sourced LLMs. For all the baselines, we report their performance on three representative backbones in Table 1, *i.e.*, LLaMA-2-7B Touvron et al. (2023), Vicuna-13B Zheng et al. (2024) and Qwen3-14B Team (2025). Additionally, we validate the effectiveness of GTool on other LLMs. Due to space limitations, detailed results are provided in Appendix B.1.

**Evaluation Metrics.** We utilize three metrics for experimental evaluation: Node F1-Score (n-F1), Link F1-Score (l-F1), and Normalized Edit Distance (NED). The n-F1 checks whether the generated plans select the right tools and l-F1 tests whether the plans preserve the topological information of the tool dependency graph. Furthermore, NED is employed to assess the correctness of the invoking order. Note the performance metrics focus primarily on the quality of the overall tool plan rather than the correctness of individual tool calls, whereas GTool's strength lies in its ability to generate coherent and effective tool plans by leveraging tool dependency modeling.

**Experiment Settings** All experiments are conducted on two NVIDIA A100-80G GPUs with CUDA compatibility. The total GPU occupation of all experiments is about 200 hours. The key hyperparameters of our model include: the number of layers in the graph neural network $n_l$, the number of edge pairs utilized during graph completion $\alpha$, the proportion of edges masked during the graph completion process $\rho$, and the weighting coefficients for the total loss computation $\lambda$. Based on extensive parameter experimentation and considering model efficiency, we empirically established the

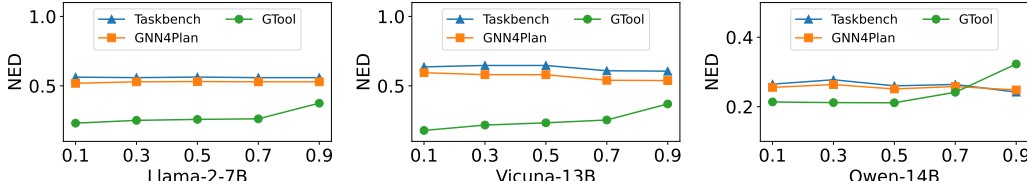

Figure 4: Performance comparison with different missing ratio of tool dependency graph.

optimal parameter configuration that demonstrates superior computational efficiency: (i) $n_l = 3$, (ii) $\alpha = 4$, (iii) $\rho = 0.1$, (iv) $\lambda = 0.1$. For all baseline methods, we have maintained their default configurations to ensure a fair and consistent comparison. In the experiments, we employed TransformerConvShi et al. (2020) as the graph neural network architecture.

## 5 EXPERIMENTAL RESULTS

We conduct extensive experiments to evaluate the performance and effectiveness of `GTool`. Due to the space limit, we only present the main results, performance on Large-scale toolset, performance of incomplete dependencies, effective experiments and ablation studies. Detailed results and analyses on missing dependency prediction, hyperparameter studies, and case studies can be found in Appendix B.3, Appendix B.4, and Appendix C, respectively.

### 5.1 OVERALL PERFORMANCE COMPARISON

The performances of baselines and `GTool` are presented in Table 1. According to the results, the following observations can be made: (1) **Performance of `GTool`.** `GTool` outperforms all baselines across almost all datasets and LLMs. The performance improvements are remarkable, *e.g.*, 26.7% growth on n-F1 compared with GNN4Plan. The superior performance of `GTool` can be attributed to the effective integration of graph neural networks and LLMs, which enables the model to capture the topological information of tool dependency graphs and generate optimal tool sequences. Among the baselines, GNN4Plan achieves the best performance, followed by HuggingGPT. Although the alignment module of GNN4Plan is effective in enhancing planning performance, it requires additional inference to generate textual steps, which is not required in `GTool`. HuggingGPT also performs better than other baselines, proving the effectiveness of prompt design. When using Vicuna-13B and Qwen3-14B as the backbone, HuggingGPT slightly outperforms `GTool` in l-F1 and NED on ToolE dataset, while underperforming in n-F1. This occurs because ToolE's short tool sequences (less than 3 steps) align with HuggingGPT's concise reasoning, optimizing its short-chain predictions. (2) **Different LLM backbones.** Comparing the results of different LLM backbones, the performance of LLaMA-2-7B is slightly worse than Vicuna-13B and Qwen3-14B, which indicates that the capacity of LLMs has an impact on planning performance. With different LLM backbones, the performance of `GTool` is consistently better than the compared baselines. This reveals that `GTool` is generalizable and robust. To assess generalizability, we further evaluate `GTool` on seven diverse backbone models. The results show that `GTool` maintains strong and stable performance across different model architectures. Full results are presented in Appendix B.1 due to space constraints. (3) **Different datasets.** For different datasets, the results on the ToolE dataset are slightly worse than the TaskBench dataset. The tools in ToolE are obtained in different domains, suggesting that the complexity of tools has an impact on the planning performance. Nevertheless, `GTool` achieves better performance, 11.7% n-F1 improvement, compared with other baselines on the ToolE dataset. (4) **Different metrics.** The performance of `GTool` is consistent across different metrics, demonstrating its effectiveness in selecting appropriate tools, generating optimal usage sequences, and preserving the topological information of tool dependency graphs. The performance on NED is slightly worse than on n-F1 and l-F1, suggesting that the correctness of invoking order is more challenging than selecting appropriate tools and generating optimal usage sequences.

### 5.2 PERFORMANCE ON LARGE-SCALE TOOLSET

To evaluate the scalability of `GTool`, we introduce the ToolBench dataset, which contains over 16,000 RESTful APIs along with tool invocation chains generated by ChatGPT. In this extended

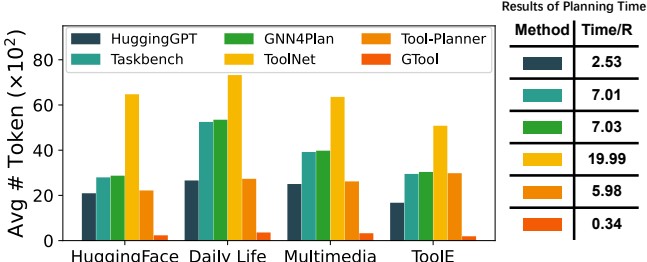

Figure 5: Efficiency results. Time is measured in seconds.

setting, we use Qwen3-14B as the backbone model and compare `GTool` with representative tuning-free and tuning-based baselines. Detailed experimental configurations are provided in Appendix A.4.

| Method | ToolBench | | | |
|---|---|---|---|---|
| | n-F1 ↑ | l-F1 ↑ | NED ↓ | Time/R↓ |
| TaskBench | 0.3649 | 0.1499 | 0.6036 | 47.24 |
| HuggingGPT | 0.5989 | 0.2290 | **0.5300** | 12.77 |
| GNN4Plan | 0.3274 | 0.2757 | 0.6773 | 47.58 |
| Tool-Planner | 0.5288 | 0.2222 | 0.5333 | 6.67 |
| GTool | **0.6126** | **0.3018** | 0.5412 | **2.02** |

Table 2: The planning performance comparison on ToolBench. Time is measured in seconds.

The results are presented in Table 2. Compared with existing baselines, `GTool` achieves an improvement of 2.28% in n-F1 and 9.46% in l-F1 under scenarios with a larger number of tools and more complex dependency structures. These results demonstrate that `GTool` maintains robust performance as the scale of the toolset increases.

In terms of efficiency, `GTool` also outperforms the baselines on large-scale datasets. This efficiency gain can be attributed to the structural characteristics of the tool graph: unlike general graphs, each tool typically depends on only a limited number of other tools. As a result, the number of tool edges grows approximately linearly with the number of tools.

### 5.3 PERFORMANCE WITH INCOMPLETE GRAPH

To verify the performance of `GTool` in incomplete tool dependency graphs, we compare `GTool` with two tuning-based methods and report the performance with different missing ratios of tool dependency graphs in Figure 4. As shown, the performance of `GTool` is consistently better than baselines across different missing ratios, evidencing that `GTool` is robust to incomplete tool dependency graphs. The performance of `GTool` is slightly worse with the increase of missing ratio. It proves that the completeness of tool dependency graphs has an impact on the planning performance. When the missing ratio reaches 90%, `GTool` underperforms the baselines on the Qwen3-14B model. This is because, under such extreme sparsity, the tool dependency information is severely missing, leaving `GTool` to rely solely on the input prompt for reasoning. In contrast to the baselines, `GTool` uses a relatively simple prompt that includes only tool names and brief instructions, which is insufficient for effective reasoning. More comprehensive results are provided in Appendix B.2.

### 5.4 EFFICIENCY STUDIES

During model inference, the computational cost of `GTool` consists of two folds: (1) the token consumption of LLMs, and (2) the time consumption of graph neural networks. Regarding token efficiency, Figure 5 presents the per-request token consumption of our method in comparison with other LLM-based approaches. As illustrated, `GTool` achieves an over 80% reduction in token consumption compared to HuggingGPT and Taskbench. The reason is that `GTool` employs request-tool graph to encode the tool descriptions instead of integrating them in prompt. For the time consumption of GNN, we compare the inference time of `GTool` with other baselines. As shown, `GTool` achieves approximately one-tenth the inference time, demonstrating the high efficiency of `GTool`.

| Backbone | Llama-2-7B | | | Vicuna-13B | | |
|---|---|---|---|---|---|---|
| | n-F1 ↑ | l-F1 ↑ | NED ↓ | n-F1 ↑ | l-F1 ↑ | NED ↓ |
| w/o All | 0.1566 | 0.0243 | 0.8611 | 0.1626 | 0.0394 | 0.8442 |
| w/o Both | 0.6131 | 0.3469 | 0.4072 | 0.7370 | 0.4831 | 0.2762 |
| w/o RS | 0.7128 | 0.4433 | 0.3108 | 0.7589 | 0.5103 | 0.2561 |
| w/o MDPL | 0.7650 | 0.5196 | 0.2541 | 0.7707 | 0.5229 | 0.2474 |
| w/ LLMlp | 0.7602 | 0.4869 | 0.2676 | 0.7784 | 0.5283 | 0.2676 |
| GTool | **0.7913** | **0.5403** | **0.2537** | **0.8029** | **0.5816** | **0.2153** |

Table 3: Ablation results of `GTool`.

## 5.5 ABLATION STUDIES

In this section, we compare `GTool` with four ablations, *i.e.*, **w/o All**, **w/o Both**, **w/o RS**, **w/o MDPL**. **w/o RS** and **w/o MDPL** remove the proposed request-specific node and missing dependency prediction with LLMs respectively. **w/o Both** removes both techniques but undergoes a training process to obtain the graph token for completing tool graphs. **w/o All** removes the tool graph and uses only the instruction and tool names as input to the language model. w/ LLMlp denotes that we remove the MDPL module and predict the tool dependencies with LLMs directly. For detailed experimental design, please refer to Appendix A.3.

As shown in Table 3, without the request-specific node, the performance decreases significantly, *e.g.* 9.92% in n-F1, 17.9% in l-F1 and 22.5% in NED on Llama-2-7B backbone. When the MDPL strategy is omitted, `GTool` exhibits a decrease of 3.32% in n-F1 and 3.83% in l-F1 and an increase of 0.15% in NED on Llama-2-7B. Replacing the MDPL strategy with a LLM, Replacing the MDPL module with a large language model led to a similar performance decline. This demonstrates that tool dependency completion based directly on an LLM fails to match the effectiveness of the dedicated MDPL module. When both modules are simultaneously ablated, the n-F1 decreases by 22.5%, the l-F1 score drops by 35.7%, and the NED increases by 60.5%. Note that **w/o Both** still achieves a performance significantly higher than that of the baseline. The graph token in **w/o Both** contains sufficient information to enhance the performance of tool learning. The results of **w/o All** are very poor. This is because `GTool` employs a very simple prompt template which only contains the tool name and a brief instruction. These results demonstrate the effectiveness of the request-specific node and the missing dependency prediction with LLMs.

When replacing the base model with Vicuna-13B in ablation studies, performance still dropped, though to a lesser extent than with LLaMA-2-7B. This is likely due to Vicuna-13B's larger size and greater robustness, which help mitigate architectural changes. These results suggest that `GTool` yields larger improvements on weaker base models.

## 5.6 ROBUSTNESS STUDY

To assess the robustness of our approach, we designed and performed **cold-start** and **transfer learning** experiments. Cold-start experiments test the model's performance in an extreme setting with no historical trajectories, where edges in the graph are entirely absent, necessitating reliance on large language models for edge completion. Transfer experiments evaluate the model's generalization ability through applying it to unseen datasets for tool planning.

The results of cold-start experiments are presented in Table 4. The experimental findings demonstrate that the performance degradation in cold-start settings varies across different large language models. Models with greater capability, *i.e.*, Vicuna-13B, are demonstrably more successful at inferring dependency relationships from the tool descriptions within the dataset, whereas less capable models struggle considerably. Moreover, the quality of the dataset itself exerts a considerable influence on the performance. Clearer descriptions in datasets substantially aid the inference process, leading to a smaller performance drop in cold-start scenarios. For instance, the tool description for ToolE is overly simplistic, leading to a more significant performance drop on this dataset compared to the Daily life dataset.

In the transfer learning experiments, `GTool` was trained on the HuggingFace and Multimedia datasets, respectively, and then directly transferred to other datasets without any fine-tuning. The experimental results are presented in Table 5. Although models trained on both datasets exhibit performance degradation when transferred to other datasets, the model trained on the HuggingFace dataset demonstrates superior generalization capability compared to the one trained on the Multime-

| Dataset | Llama-2-7B | | | Vicuna-13B | | |
|---------|------------|--------|-------|------------|--------|-------|
| | n-F1 ↑ | l-F1 ↑ | NED ↓ | n-F1 ↑ | l-F1 ↑ | NED ↓ |
| HuggingFace | 0.7374 | 0.4684 | 0.2855 | 0.7649 | 0.5225 | 0.2538 |
| Daily life | 0.9103 | 0.7711 | 0.1106 | 0.9808 | 0.8835 | 0.0477 |
| Multimedia | 0.7789 | 0.5453 | 0.2454 | 0.7910 | 0.5743 | 0.2286 |
| ToolE | 0.7180 | 0.3200 | 0.3850 | 0.6955 | 0.3044 | 0.3926 |

Table 4: Results of cold-start experiments.

| Dataset | Pretrained on HuggingFace | | | Pretrained on Multimedia | | |
|---------|---------------------------|--------|-------|--------------------------|--------|-------|
| | n-F1 ↑ | l-F1 ↑ | NED ↓ | n-F1 ↑ | l-F1 ↑ | NED ↓ |
| HuggingFace | - | - | - | 0.5934 | 0.2835 | 0.4245 |
| Daily life | 0.8479 | 0.6699 | 0.1831 | 0.8365 | 0.6538 | 0.2005 |
| Multimedia | 0.6964 | 0.4511 | 0.3303 | - | - | - |
| ToolE | 0.7252 | 0.3366 | 0.3445 | 0.7049 | 0.3091 | 0.3785 |

Table 5: Results of transfer learning experiments.

dia dataset. This can be attributed to the greater diversity and richness of tool types in HuggingFace, whereas Multimedia is confined to tools related to multimedia applications. This finding suggests a promising direction: large-scale tool-based pre-training may endow `GTool` with stronger generalization abilities.

## 6 RELATED WORKS

**Tool Learning.** Current tool learning works Qin et al. (2024a); Qu et al. (2025) enable foundation models to use tools like humans, involving tool planning Qin et al. (2024b); Liu et al. (2024c) and parameter completion Hao et al. (2023); Wang et al. (2024a). Advances include training paradigms Park et al. (2023); Liu et al. (2024b) and generalization strategies Qin et al. (2024a); Gao et al. (2024). Tool planning focuses on selecting and sequencing tools Qin et al. (2024b); Yuan et al. (2024); Liu et al. (2024a) or task decomposition Liang et al. (2024); Qian et al. (2023a); Kong et al. (2024); Liu et al. (2024c). Parameter completion uses sequence-to-sequence models or structured prediction to map instructions to executable parameters Yang et al. (2024b); Hao et al. (2023); Wang et al. (2024a). Recent advancements include supervised learning, trial-and-error, and graph-informed supervision Park et al. (2023); Qian et al. (2023b); Liu et al. (2024b), as well as meta and curriculum tool learning Qin et al. (2024a); Gao et al. (2024). However, these efforts often neglect tool trajectory information and struggle with incomplete tool interaction knowledge.

**Large Language Models on Graphs.** Prior work integrates LLMs with graph-structured knowledge Jin et al. (2024), enhancing planning tasks. Approaches include graph as sequence Tian et al. (2024); Huang et al. (2024b), graph-empowered LLM Jin et al. (2023), and graph-aware LLM fine-tuning Zhu et al. (2024). Graph as sequence methods encode graph structures into sequential inputs Ye et al. (2023); Tang et al. (2024), but suffer from structural information loss and increased computational burden. Graph-empowered LLMs modify Transformer architectures to encode text and graphs via hybrid attention mechanisms Zhang et al. (2022); Jin et al. (2023), but face high adaptation costs. Graph-aware fine-tuning injects graph knowledge by fine-tuning LLMs on graph-derived objectives Zhu et al. (2024), but relies on complete underlying graphs, which is often unmet in real-world tool ecosystems. In addition, massive graph mining works Kipf & Welling (2016); Zhang & Chen (2018); Chami et al. (2019) learn to extract node representations for estimating missing links of graph. Without considering the modality gap, these works cannot directly used in tool planning.

## 7 CONCLUSION

`GTool` is proposed to improve the performance of tool planning by integrating tool dependencies. It employs missing edge prediction to enhance the reliability of incomplete tool dependency scenarios. The user requests are integrated into the dependency graph for efficient tool planning. Extensive experiments demonstrate that `GTool` not only achieves state-of-the-art performance but also reduces the planning time significantly. Moreover, `GTool` is robust to missing dependencies and can be easily generalized to different LLM backbones. In the future, we plan to evaluate the effectiveness of `GTool` on more powerful LLMs and its scalability to large-scale tool collections. Moreover, we plan to explore more advanced techniques to improve the quality of the tool graph, such as retrieval-augmented generation and reinforcement learning.

ACKNOWLEDGMENTS

This work was supported by the National Natural Science Foundation of China (NSFC) under Grant Nos. 62472405 and 62372429, and by the Strategic Priority Research Program of the Chinese Academy of Sciences under Grant No. XDB0680101. This work was also partially supported by Alibaba Innovative Research.

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

APPENDIX

# A EXPERIMENT CONFIGURATIONS

In this section, we provide detailed information on the experimental configurations, including the details of baselines and ablation settings.

## A.1 DETAILS OF DATASETS

Detailed information regarding these four datasets is presented in Table 6.

Table 6: Statistics of datasets.

| Dataset | $\#_{tool}$ | $\#_{edge}$ | $\#_{train}$ | $\#_{vail}$ | $\#_{test}$ |
|---------|------|------|-------|------|------|
| HuggingFace | 23 | 225 | 2178 | 726 | 726 |
| Multimedia | 40 | 449 | 1788 | 596 | 597 |
| Daily Life | 40 | 1560 | 1672 | 57 | 558 |
| ToolE | 15 | 154 | 298 | 99 | 100 |

We further analyze the number of tools required per request within the dataset, and the distribution is shown in Figure 6(a). It can be observed that cases requiring more than two tools are fairly common.

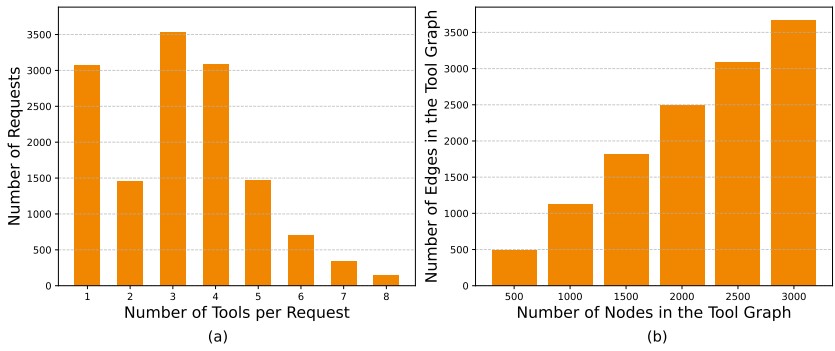

Figure 6: Statistics of dataset characteristics. (a) shows the distribution of the number of requests grouped by the number of tools involved in each request. (b) shows the distribution of the number of nodes and edges in tool graphs.

## A.2 DETAILS OF BASELINES

Eight baselines are implemented in Section 5.1 with the following technical specifications:

**BM25.** Employed as our lexical retrieval baseline, this probabilistic model calculates query-document relevance scores through term frequency-inverse document frequency (TF-IDF) statistical analysis. The top-5 most relevant documents are selected based on cosine similarity ranking to construct action plans.

**COLT.** This approach enhances tool planning by establishing dual bipartite graphs to retrieve optimal tools: 1) request-scenario graph and 2) scenario-tool graph. We utilize GPT-4o for automated scenario generation. The top-5 most similar tools are aggregated through similarity sorting.

**Graph RAG-Tool Fusion.** This method combines vector-based retrieval with graph traversal to efficiently construct a tool planning chain from a predefined tool dependency graph. In our experiments, we set the top-k value for similarity-based retrieval to 3, and constrain the depth of the depth-first search to a maximum of 3.

**HuggingGPT.** This method adopts a sophisticated chain-of-thought prompting strategy for multi-stage task planning. The number of few-shot demonstrations is set to 1, following the default configuration of the original article.

**TaskBench.** We adopt the authors' proposed template engineering methodology for cross-LLM evaluation. To ensure fair comparison, we maintain identical few-shot settings (k=1) across all baseline implementations.

**GNN4Plan.** This graph-enhanced planner addresses the biases of LLMs, such as attention and auto-regressive loss, in structured reasoning using graph neural networks. We freeze the backbone language model parameters and choose SGC as GNN encoder.

**ToolNet.** This work structures tools as a directed graph, where each node represents a tool and weighted edges indicate possible transitions between them. Starting from an initial node, a large language model navigates the graph by iteratively selecting the next tool from the successors, continuing this process until the task is completed.

**Tool-Planner.** Tool-Planner groups tools with the same function into a toolkit and allows LLMs to implement planning across the various toolkits. The number of toolkit is set to 10 during the experimental phase to evaluate the model's performance.

| Backbone | HuggingFace | | | Daily Life | | |
|---|---|---|---|---|---|---|
| | n-F1 ↑ | l-F1 ↑ | NED ↓ | n-F1 ↑ | l-F1 ↑ | NED ↓ |
| Llama-2-13B | 0.7933 | 0.5685 | 0.2244 | 0.9657 | 0.8718 | 0.0578 |
| CodeLlama-13B-hf | 0.8082 | 0.5895 | 0.2082 | 0.9631 | 0.8683 | 0.0598 |
| DeepSeek-R1-Distill-Llama-8B | 0.7706 | 0.5321 | 0.2439 | 0.9637 | 0.8545 | 0.0635 |
| Mistral-7B-v0.1 | 0.8043 | 0.5834 | 0.2107 | 0.9656 | 0.8717 | 0.0578 |
| Qwen2-7B | 0.7792 | 0.5542 | 0.2367 | 0.9620 | 0.8643 | 0.0633 |
| Vicuna-7B | 0.7616 | 0.4987 | 0.2645 | 0.9274 | 0.8097 | 0.0976 |
| Yi-6B | 0.7465 | 0.4749 | 0.2812 | 0.9305 | 0.8003 | 0.0913 |

Table 7: Result of extended model architecture experiment on HuggingFace and Daily Life datasets.

| Backbone | Multimedia | | | ToolE | | |
|---|---|---|---|---|---|---|
| | n-F1 ↑ | l-F1 ↑ | NED ↓ | n-F1 ↑ | l-F1 ↑ | NED ↓ |
| Llama-2-13B | 0.8154 | 0.6247 | 0.2058 | 0.7550 | 0.320 | 0.365 |
| CodeLlama-13B-hf | 0.8506 | 0.6765 | 0.1689 | 0.7150 | 0.3200 | 0.3650 |
| DeepSeek-R1-Distill-Llama-8B | 0.8024 | 0.5758 | 0.2203 | 0.6631 | 0.2400 | 0.4435 |
| Mistral-7B-v0.1 | 0.8126 | 0.6201 | 0.2099 | 0.7586 | 0.3522 | 0.3514 |
| Qwen2-7B | 0.8069 | 0.6108 | 0.2126 | 0.7043 | 0.3382 | 0.3797 |
| Vicuna-7B | 0.7320 | 0.4674 | 0.2927 | 0.6582 | 0.2533 | 0.4361 |
| Yi-6B | 0.7558 | 0.5054 | 0.2737 | 0.6990 | 0.2933 | 0.3950 |

Table 8: Result of extended model architecture experiment on Multimedia and ToolE datasets.

## A.3 ABLATION SETTINGS

To validate the effectiveness of our design, we conduct the following four ablation studies:

- Without request-specific node (w/o RS Node): Here, we remove the request-specific node from the tool graph. Instead of utilizing the request-specific node's embedding, we compute the graph vector by applying average pooling to the embeddings of all nodes. This experiment is designed to assess the significance of the request-specific node in capturing graph-level information.

- Without missing dependency prediction with LLMs (w/o MDPL): In this scenario, we omit the optimization step described in Section 3.2.2. The tool graph obtained directly from Section 3.1.2 is used for training without any further refinement.

- Without request-specific node and missing dependency prediction with LLMs (w/o Both): The removal of both the RS Node and the MDPL process is conducted following the same methodology as described above.

- Without all additional modules (w/o All): We conduct inference without encoding any graph information, relying solely on the prompt that does not include graph tokens.

### A.4 EXPERIMENTS ON A LARGER-SCALE DATASET

This experiment is conducted on the ToolBench dataset, which covers multiple instruction scenarios, including single-tool instructions (I1), intra-category multi-tool instructions (I2), and intra-collection multi-tool instructions (I3). Since the I1 scenario involves only single-tool usage, it does not align with `GTool`'s multi-tool planning setting and is therefore excluded from our evaluation.

For the I2 and I3 scenarios, we follow the original paper's retrieval strategy to select relevant tool nodes. Low-quality tools with inaccurate or missing descriptions are filtered out. Based on the existing tool invocation chains in the dataset, we construct tool dependency graphs by extracting directed edges among the retrieved tools.

During the graph construction process, we observe that the number of edges grows approximately linearly with the number of nodes. The detailed statistics are shown in Figure 6(b). This linear growth reflects the bounded nature of tool dependencies—each tool typically depends on only a few others. While this sparsity is less apparent in small-scale datasets such as TaskBench and ToolE, it becomes increasingly significant as the tool set grows, resulting in progressively sparser tool graphs.

## B MORE EXPERIMENTAL RESULTS

In this section, we provide additional experimental results to further validate the effectiveness of `GTool`. Specifically, we present the performance of `GTool` on more LLM backbones and provide a comprehensive robustness analysis across extended evaluation metrics.

### B.1 PERFORMANCE ON MORE LLM BACKBONES

To systematically validate the cross-architectural robustness of `GTool`, we conduct comprehensive ablation studies across 6 foundational language models ranging from 7B to 13B parameters, including Llama-2-13BTouvron et al. (2023), CodeLlama-13B-hfRoziere et al. (2023), Mistral-7B-v0.1Jiang et al. (2023), Qwen2-7BYang et al. (2024a), Vicuna-7BZheng et al. (2024) and Yi-6B. As evidenced in Table 7 and Table. 8, `GTool` demonstrates remarkable consistency with 0.081 performance standard deviation across heterogeneous model architectures, while maintaining progressive scaling characteristics. Three critical observations emerge:

(1) Consistent with neural scaling laws, `GTool` demonstrates significant performance gains when scaling from 6B to 13B model sizes, as measured by our unified evaluation protocol.

(2)Notably, Mistral-7B-v0.1 outperforms LLaMA-2-13B across all evaluation metrics, despite having 46% fewer parameters, which can be attributed to its optimized architectural design.

### B.2 PERFORMANCE OF OTHER METRICS WITH INCOMPLETE GRAPH

Due to space constraints in the main manuscript, Section 5.3 primarily presents the NED evaluation metric outcomes. For comprehensive documentation, we hereby provide complete experimental results across all evaluation metrics in Figure 7.

Consistent with the n-F1 evaluation pattern, experimental results on n-F1 and l-F1 metrics demonstrate `GTool`'s robust performance characteristics.

### B.3 ACCURACY OF MISSING DEPENDENCY PREDICTION

To quantitatively evaluate the missing dependency prediction approach, we conduct link prediction experiments using the edge set $S$ defined in Equation 2 with frozen trained models.

As shown on Table 9, the experimental results obtained from HuggingFace datasets demonstrate that all three base models achieve prediction accuracy rates exceeding 82%, which validates the model's effectiveness in predicting dependency relationships among tools.

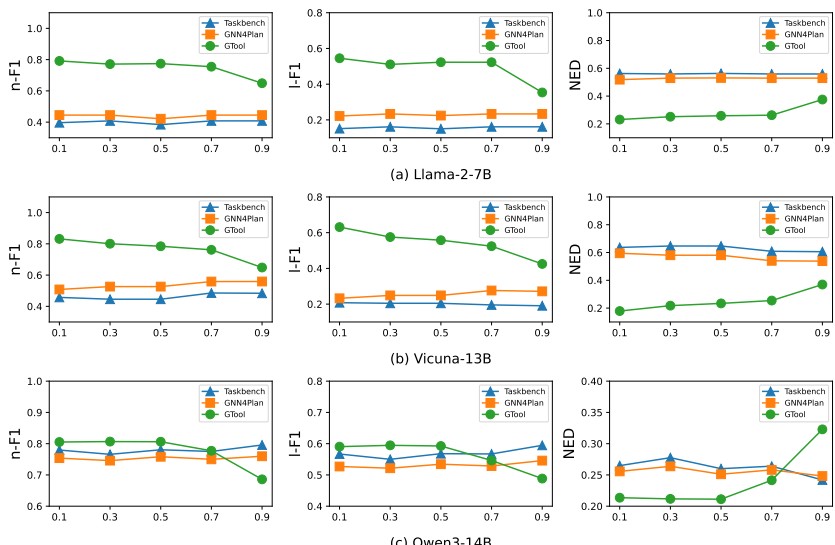

Figure 7: Performance comparison with different mask ratio with extended metrics: n-F1, l-F1 and NED.

| Backbone | Accuracy |
|---|---|
| Llama-2-7B | 0.8471 |
| Vicuna-13B | 0.8529 |
| Qwen-14B | 0.8221 |

Table 9: Experimental evaluation of missing dependency prediction accuracy.

### B.4 HYPERPARAMETER ANALYSIS

GTool introduces three critical hyperparameters: the sample size of missing dependency prediction with LLMs $\alpha$, balance factor $\lambda$ and the number of GNN layers $n_l$. To investigate their impacts, we test the performance of GTool under different settings. Figure 8 presents the experimental results, based on which we make the following observations.

**Influence of $\alpha$.** According to the results, $\alpha$ exhibits diminishing returns beyond $\alpha$=4. While larger $\alpha$ values marginally improve representation diversity (0.13% gain from $\alpha$=4 to $\alpha$=6). The performance-stability trade-off analysis justifies our selection of $\alpha$=4 as the best configuration.

**Influence of $\lambda$.** The performance demonstrates a distinct bell-shaped curve relative to $\lambda$ values, peaking at $\lambda$=0.1 before subsequent degradation. This non-monotonic relationship suggests that moderate regularization strength through $\lambda$ effectively balances model capacity and generalization. We find $\lambda$=0.1 is the optimal configuration.

**Influence of $n_l$.** With the increase of $n_l$, the performance of GTool initially improves and then declines. It proves that GTool suffers from overfitting problem when $n_l$ is large. In our experiments, we choose $n_l = 3$ which is a suitable setting for driving optimal planning performance.

## C CASE STUDIES

This section presents visualizations of representative dataset instances and examines how tool dependencies influence tool learning, as reflected in the model's outputs.

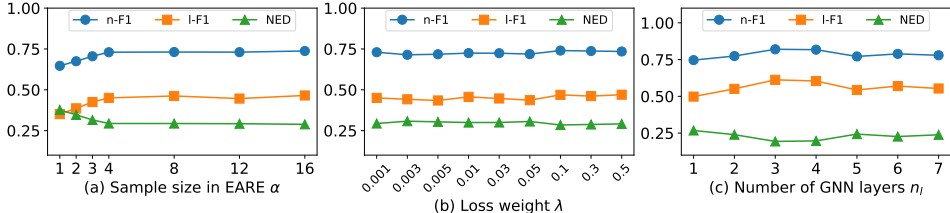

Figure 8: Results of hyperparameter analysis: (a) the effect of the sample size in MDPL $\alpha$ ; (b) the influence of loss weight $\lambda$; (3) the impact of the number of GNN layers $n_l$.

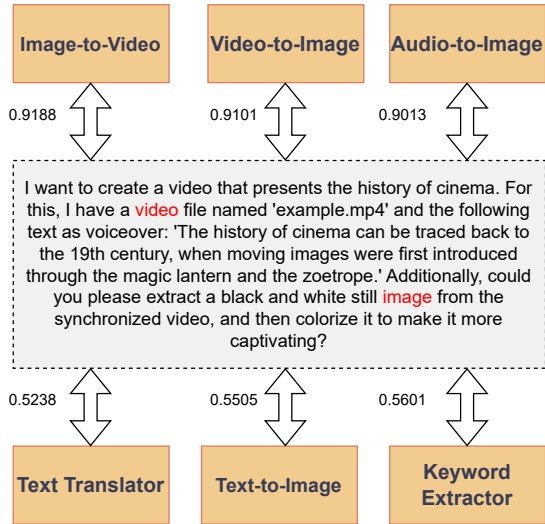

Figure 9: Cosine similarities between the Graph Token and nodes in the tool dependency graph.

### C.1 SEMANTIC ANALYSIS OF THE GRAPH TOKEN

To further examine the semantics encoded in the Graph Token, we compute the cosine similarity between the Graph Token representation and the node embeddings in the trained tool-dependency graph. As shown in Figure 9, in the given request, the user asks to extract an image from a video based on its content. The three nodes with the highest cosine similarity are "Image-to-Video," "Video-to-Image," and "Audio-to-Image," while the three least similar nodes are "Text Translator," "Text-to-Image," and "Keyword Extractor." These results suggest that the Graph Token indeed captures meaningful semantic information that aligns closely with the user's intent.

### C.2 VISUAL CASE ANALYSIS OF TOOL DEPENDENCY

Tool dependency, as a topology defined over tools, has a unique characteristic. It is highly task-dependent. In the absence of a specific task, such dependencies remain implicit. When tools exist independently, they can be invoked separately without any ordering constraints. However, once a task is specified as context, a topological structure emerges among the tools. As illustrated in Figure 10, there exists an implicit dependency graph among the three tools shown. Request 1 and Request 2 involve similar tasks, but Request 2 includes an additional requirement to enhance the audio. As a result, the input to the speech classification tool becomes dependent on the output of the speech-to-speech tool. Consequently, the tool trajectory changes due to the altered task context.

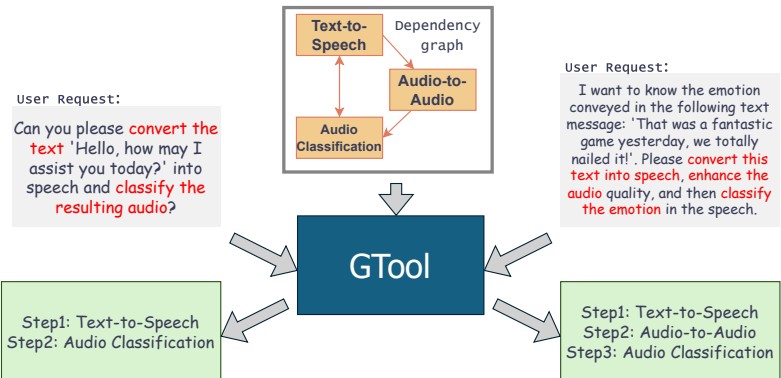

Figure 10: The figure illustrates two similar requests and their respective tool planning outcomes, along with the corresponding tool dependency graphs. A change in the request leads to a shift in tool dependencies, ultimately resulting in different planning trajectories.

## C.3 VISUAL CASE ANALYSIS OF USER REQUEST

The user's request is equally crucial for tool-planning outcomes. Even with identical tool dependencies, slight differences in user instructions can lead to entirely different plans. As shown in Figure 11, although the dependency graphs are exactly the same, User 1 requires "classification before translation," whereas User 2 requests "translation before classification," resulting in distinct planning outputs.

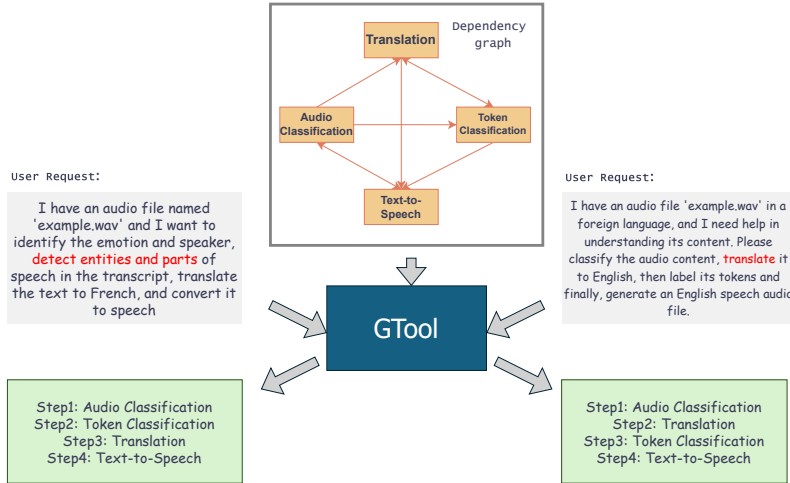

Figure 11: The figure shows two user requests that share identical tool dependency graphs but yield different tool-planning results. Although the dependencies remain unchanged, subtle differences in the user instructions—requesting classification before translation versus translation before classification—lead to distinct planning trajectories.

## C.4 VISUAL CASE ANALYSIS OF TOOL PLANNING OUTPUTS

Through comparative case studies in Table 10, Table 11 and Table 12, we dissect planning outcomes from top-performing baselines (BM25, TaskBench, HuggingGPT) and `GTool`, revealing critical methodological limitations:

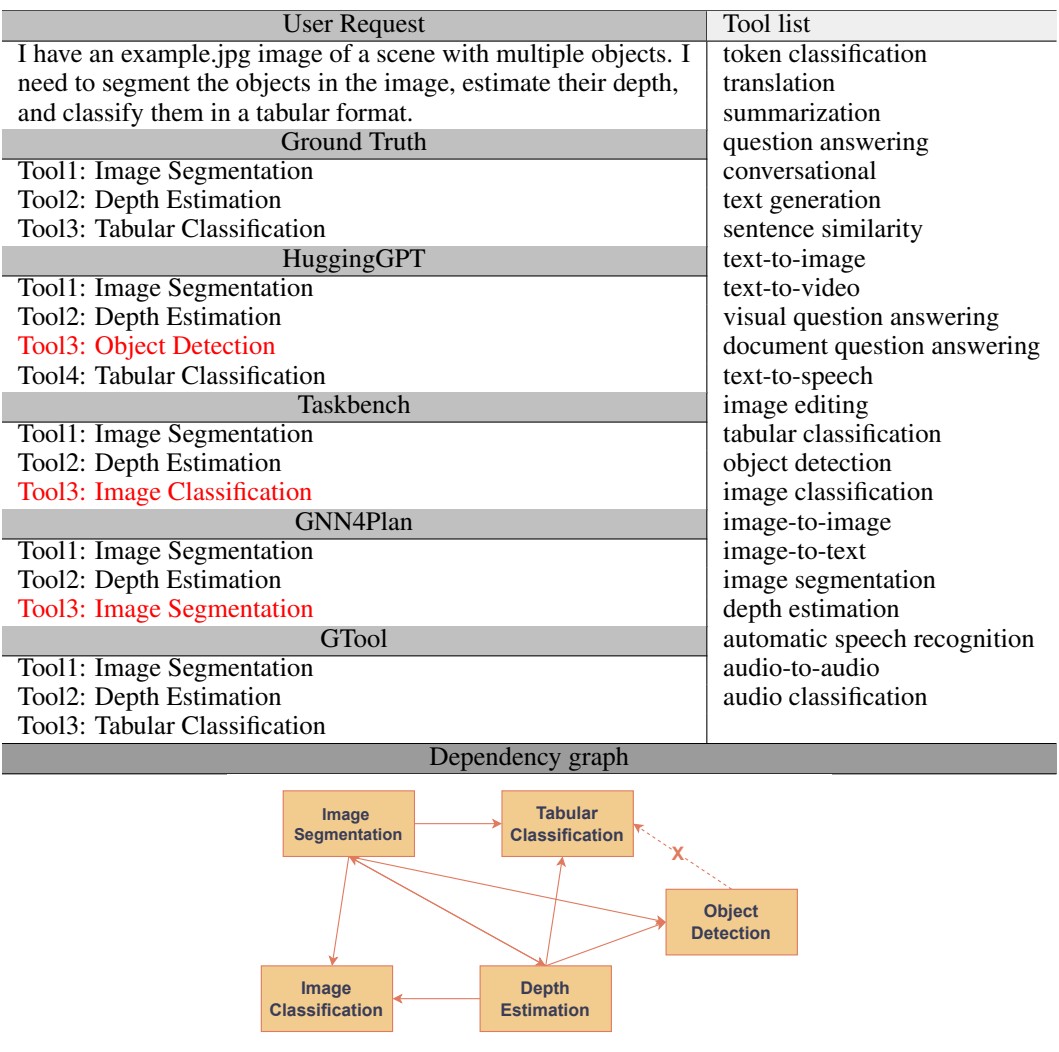

| User Request | Tool list |
|---|---|
| I have an example.jpg image of a scene with multiple objects. I need to segment the objects in the image, estimate their depth, and classify them in a tabular format. | token classification |
| | translation |
| | summarization |
| **Ground Truth** | question answering |
| Tool1: Image Segmentation | conversational |
| Tool2: Depth Estimation | text generation |
| Tool3: Tabular Classification | sentence similarity |
| **HuggingGPT** | text-to-image |
| Tool1: Image Segmentation | text-to-video |
| Tool2: Depth Estimation | visual question answering |
| Tool3: Object Detection | document question answering |
| Tool4: Tabular Classification | text-to-speech |
| **Taskbench** | image editing |
| Tool1: Image Segmentation | tabular classification |
| Tool2: Depth Estimation | object detection |
| Tool3: Image Classification | image classification |
| **GNN4Plan** | image-to-image |
| Tool1: Image Segmentation | image-to-text |
| Tool2: Depth Estimation | image segmentation |
| Tool3: Image Segmentation | depth estimation |
| **GTool** | automatic speech recognition |
| Tool1: Image Segmentation | audio-to-audio |
| Tool2: Depth Estimation | audio classification |
| Tool3: Tabular Classification | |
| **Dependency graph** | |

Table 10: The figure demonstrates a concrete case study comprising the original user request, the generated planning sequence and the corresponding tool dependency graph. Erroneous planning steps are annotated in red, while missing dependencies are explicitly denoted by dashed lines with cross markers.

**Dependency Neglect**. Table 10 illustrates a representative case study comprising a user request, planning outcomes from baseline methods, and their corresponding tool dependency graphs. The results reveal that HuggingGPT's planning sequence incorporates an additional "image segmentation" step. Although this operation demonstrates semantic relevance to the user's objective, the tool graph exhibits a critical structural deficiency—the absence of a directed dependency edge from "image segmentation" to "text classification". This observation indicates that the model prioritizes semantic associations while neglecting mandatory tool dependencies, thereby violating workflow integrity constraints.

**Precision deficiency.** As demonstrated in Table 11, all baseline methods erroneously initiate with "Image-to-Text" despite ground truth requiring "Visual Question Answering". The observed errors stem from the semantic proximity between tool functionalities, which exceeds baseline methods' discrimination capabilities.

**Limitations in processing complex dependency information.** This limitation is empirically evidenced by the erroneous tool edges generated in TaskBench (Table 12), where even explicit textual input of complete graph structures fails to guarantee accurate reasoning. Such observations sug-

| User Request | Tool list |
|---|---|
| Convert the following text: 'Rainy clouds are forming in the sky.' into an audio file, enhance the audio quality, then transcribe it back to text. Use the transcribed text to change example.jpg to a more relevant image. Answer this question based on the edited image: 'What is the weather like in the image?'. Finally, generate a video based on the answer. | token classification |
| | translation |
| | summarization |
| | question answering |
| | conversational |
| | text generation |
| **Ground Truth** | sentence similarity |
| Tool1: Text-to-Speech | text-to-image |
| Tool2: Audio-to-Audio | text-to-video |
| Tool3: Automatic Speech Recognition | visual question answering |
| Tool4: Image Editing | document question answering |
| Tool5: Visual Question Answering | text-to-speech |
| Tool6: Text-to-Video | image editing |
| **HuggingGPT** | tabular classification |
| Tool1: Text-to-Speech | object detection |
| Tool2: Image Classification | image classification |
| **TaskBench** | image-to-image |
| Tool1: Text-to-Speech | image-to-text |
| Tool2: Audio-to-Audio | image segmentation |
| Tool3: Automatic Speech Recognition | depth estimation |
| Tool4: Image-to-Image | automatic speech recognition |
| Tool5: Visual Question Answering | audio-to-audio |
| **GNN4Plan** | audio classification |
| Tool1: Text-to-Speech | |
| Tool2: Audio-to-Audio | |
| Tool3: Automatic Speech Recognition | |
| Tool4: Text-to-Image | |
| Tool5: Visual Question Answering | |
| **GTool** | |
| Tool1: Text-to-Speech | |
| Tool2: Audio-to-Audio | |
| Tool3: Automatic Speech Recognition | |
| Tool4: Image Editing | |
| Tool5: Visual Question Answering | |
| Tool6: Text-to-Video | |

Table 11: Common failure patterns in baseline methods, including incorrect tool selection (highlighted in red) and tool omission.

gest that LLMs inherently lack the capacity to parse dense topological dependencies through purely sequential text prompts, a challenge exacerbated by the models' context window limitations.

### C.5 BAD CASE ANALYSIS

Many existing tools have overlapping functionalities and similar descriptions, which poses significant challenges for GTool's planning, as illustrated in Figure 12. In this task, the correct tool is "Document Question Answering," but the presence of a similar tool, "Question Answering," leads to incorrect planning by GTool. Such cases are common in real-world scenarios, highlighting the need for mechanisms to correct planning outputs. While this is beyond the scope of the current work, we plan to address it in future research. One potential approach is to leverage multi-turn interactions with an LLM to refine the planning results.

## D THE USE OF LARGE LANGUAGE MODELS

In this work, we only used large language models for manuscript polishing and word choice refinement, as well as for checking grammar and typographical errors. We explicitly state that the

| User Request | Tool list |
|---|---|
| I have an image 'example.jpg' containing a scene of an event, and I want to know what is happening in the image. Then, generate a short related text with the answer, convert the text to speech, and classify the audio content. | token classification |
| | translation |
| | summarization |
| | question answering |
| **Ground Truth** | conversational |
| Tool1: Visual Question Answering | text generation |
| Tool2: Text Generation | sentence similarity |
| Tool3: Text-to-Speech | text-to-image |
| Tool4: Audio Classification | text-to-video |
| **HuggingGPT** | visual question answering |
| Tool1: Image-to-Text | document question answering |
| Tool2: Text Summarization | text-to-speech |
| Tool3: Text-to-Speech | image editing |
| Tool4: Audio Classification | tabular classification |
| **TaskBench** | object detection |
| Tool1: Image-to-Text | image classification |
| Tool2: Text Classification | image-to-image |
| Tool3: Text Summarization | image-to-text |
| Tool4: Text-to-Speech | image segmentation |
| Tool5: Audio Classification | depth estimation |
| **GNN4Plan** | automatic speech recognition |
| Tool1: Image-to-Text | audio-to-audio |
| Tool2: Text Generation | audio classification |
| Tool3: Summarization | |
| Tool4: Text-to-Speech | |
| Tool5: Audio Classification | |
| **GTool** | |
| Tool1: Visual Question Answering | |
| Tool2: Text Generation | |
| Tool3: Text-to-Speech | |
| Tool4: Audio Classification | |

Table 12: Common failure patterns in baseline methods, including incorrect tool selection (highlighted in red) and tool redundancy.

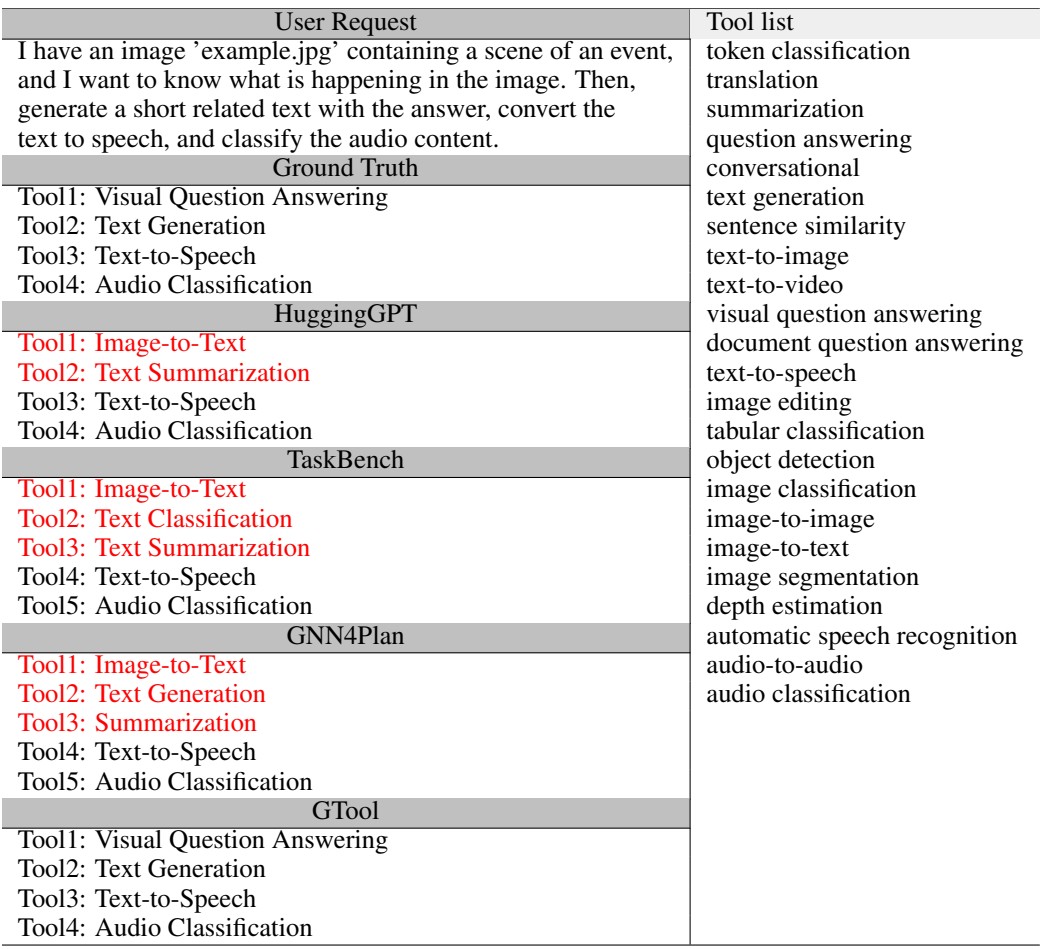

Figure 12: Bad case planned by GTool

conceptualization of the ideas, the design of the models, and the implementation of the code were entirely performed without any assistance from large language models.

