# OpenReview forum: "GTool: Graph Enhanced Tool Planning with Large Language Model"
_ICLR.cc/2026/Conference — ICLR 2026 Poster_

### Official Review · Reviewer_eanv · 2025-10-26

**Soundness:** 3
**Presentation:** 4
**Contribution:** 3
**Rating:** 4
**Confidence:** 4

**Summary:**

The paper presents GTool, a novel graph-enhanced framework for tool planning with large language models (LLMs). Existing tool-using systems (e.g., HuggingGPT, ToolNet) typically treat tools as independent modules and fail to model the latent dependency structure among them, leading to suboptimal planning. GTool addresses this limitation by explicitly encoding tool dependencies as a graph and integrating graph neural networks (GNNs) with frozen LLMs to perform structured reasoning.

**Strengths:**

1.This paper presents a very interesting perspective. Unlike traditional approaches such as ToolNet, it does not mechanically learn and plan tool invocation patterns. Instead, it systematically performs graph neural network–assisted learning based on historical tool dependencies and invocation paradigms. The theoretical foundation is strong, and the relationships between dependencies and corresponding tool-handling strategies are clearly and directly defined.

2.GTool introduces a novel direction for tool invocation schemes. While traditional methods focus more on handling the current state, this work emphasizes the dependency relationships among tools. It provides detailed analysis on potential issues in planning and decomposing a problem, especially in the formalization of multi-dependency iterative invocations. Traditional tool-calling frameworks tend to be linear, but this approach can address deeper, multi-layered problems more effectively.

3.Another strength of this paper lies in its impressive experimental results. Considering the complexity of tool invocation, the authors conducted experiments across multiple datasets and demonstrated the superiority of their method in terms of invocation time and performance. In particular, the proposed lightweight GNN architecture showed strong optimization capability compared to traditional long-CoT–based tool learning approaches.

**Weaknesses:**

1. My biggest concern is that the authors did not provide any examples or analysis of incorrect tool calls—they only examined incomplete graphs. Compared to the results on incomplete graphs, I would like to see whether the framework has the capability to adjust tool invocations after extraction through GTool. Put more simply, if we find that a certain dependency extracted by GTool fails to satisfy the basic task requirements, how does the framework adapt or adjust? Or does this paper simply not cover such adaptive capability? If the proposed method is to be applied to broader tool-use scenarios, then the ability to identify and adjust tools is even more important than fast inference.

2. Following the previous point, we know that methods such as Tool-Planner and ToolNet mainly focus on handling problematic tool calls—their performance improvements often come from repairing or re-planning calls when errors occur. If GTool merely produces a plan without further revising its correctness, then I think the reported performance gains are somewhat unfair. After all, in traditional approaches like Toolformer or DFSDT, if every tool call were correct, their performance metrics would essentially match GTool’s.

3. Could you further explain your final loss function? In what specific setting or stage is this loss jointly trained?

4. In large-scale tool scenarios, why does GTool perform better in your experiments? Do you think this is related to the graph’s inherent understanding of tool relations? Given that with 16k tools the contextual information could become excessive—leading to higher hallucination rates—I wonder whether the graph’s strong dependency modeling might also increase the overall error rate.

**Questions:**

I remain open to further discussion from the authors. If they can address these issues more thoroughly or provide additional analysis or experiments, I would be willing to raise my original score.

---

> ### Author Response · Authors · 2025-11-21
>
> **Incorrect tool calls(W1).** We appreciate the valuable feedback regarding the analysis of incorrect tool calls. In the revised manuscript, we have included additional case studies that specifically focus on instances of erroneous tool invocations. Our analysis reveals that GTool often prevents incorrect tool calls from occurring in the first place, thanks to its effective modeling of tool dependencies, which enhances the model's understanding of the relationships between tools and reduces the likelihood of errors. For cases where erroneous tool calls do occur, we discuss potential strategies for adaptation and adjustment, although this aspect is not the primary focus of the current work. We acknowledge the importance of such adaptive capabilities for broader tool-use scenarios and suggest it as a direction for future research. These additions aim to provide a more comprehensive understanding of GTool's performance in handling incorrect tool calls.
>
> **Metrics(W2).** Sorry for the misunderstanding. The performance metrics used in our evaluation focus primarily on the quality of tool planning. For methods containing tool-calling, such as Tool-Planner and ToolNet, we report their single LLM interaction planning performances to ensure fair comparison. We agree that multiple LLM interactions would repair wrong plans and this could be the future direction of our method. In the revised manuscript, the evaluation metrics are explicitly stated to make the experiments more clear.
>
> **Loss function(W3).** Thanks for the suggestion. We have provided a more detailed explanation of the loss function, including its components and how it is jointly trained during the model's training process. Specifically, the loss function combines terms that account for both the accuracy of tool dependency predictions and the quality of the generated tool plans. This joint training approach allows GTool to effectively learn from both aspects simultaneously. We have included this information in Section 4 of the revised manuscript for clarity.
>
> **Large-scale tool scenarios(W4).** Thanks for the insightful question. Firstly, in Appendix C.1, we compare the embedding of query node with tool nodes and provide representative cases to explain the semantics of the query node. Moreover, we provide the most similar word tokens based on the query node embedding and word embeddings in Appendix C.1. The results indicate that graph token in GTool represents meaningful language semantics. In our experiments, GTool demonstrates superior performance primarily due to its effective modeling of tool dependencies, which allows it to better understand and utilize the relationships between tools. This inherent understanding helps mitigate issues related to excessive contextual information that can lead to hallucination in traditional LLM-based approaches.

---

### Official Review · Reviewer_6pZ4 · 2025-10-26

**Soundness:** 3
**Presentation:** 3
**Contribution:** 3
**Rating:** 6
**Confidence:** 4

**Summary:**

This paper proposes GTool, a graph-enhanced framework for tool planning with large language models. It constructs a request-specific tool graph to model dependencies among tools, encodes this structure with a graph neural network, and injects the resulting <graph token> into the LLM to guide tool selection and ordering. A missing-dependency prediction task allows GTool to handle incomplete graphs, and only the lightweight GNN is trained while the LLM remains frozen. Experiments show that GTool improves planning accuracy by over 29%, maintains robustness under missing dependencies, and greatly reduces inference cost.

**Strengths:**

1. The paper proposes a genuinely novel and well-motivated idea — representing tool dependencies explicitly as a graph and integrating it with LLMs through a GNN encoder. This design feels natural yet surprisingly underexplored in prior work, and the authors manage to make it both elegant and technically grounded. I particularly appreciate how the paper bridges symbolic structure (graph reasoning) and LLM-based semantic planning in a coherent way.
2. One of the most impressive aspects is the robustness under incomplete or noisy dependency graphs. The “Missing Dependency Prediction” component is not only clever but also practical — it addresses a real pain point in existing methods that assume full dependency knowledge. The fact that GTool still performs well when 90% of edges are missing is very compelling.
3. I like that the model does not fine-tune the LLM itself — only the GNN encoder is trained. This is a huge advantage in terms of scalability and reproducibility. The reported reduction of 95% in token length and 10× faster inference time makes the method appealing not just conceptually but also computationally.
4. The experiments are thorough and consistent. The improvement (around +30% F1 on average) is significant and holds across multiple datasets and backbones (LLaMA, Vicuna, Qwen). The ablation studies are also well-designed, showing the clear contribution of each component.
5. The paper is well-structured and easy to follow. Figures and equations are clean, and the intuition behind each design choice (e.g., why a <graph token> is needed) is clearly articulated. The work feels mature — not just a proof of concept, but something that could realistically be integrated into production-level systems.

**Weaknesses:**

1. The paper provides basic ablations (e.g., removing <graph token> or MDPL), but it would be interesting to see more analysis of what the GNN actually learns — for example, visualizing attention weights or node embeddings to illustrate that it truly captures tool relationships rather than acting as a generic feature compressor.

2. While the writing is overall clear, the training description could be elaborated — e.g., how the <graph token> is injected into the LLM embedding space in practice, and whether this alignment requires any additional projection layer.

3. The construction of the initial tool dependency graph relies on observed invocation trajectories. In domains where such history is sparse or noisy, the method might struggle to bootstrap meaningful structure. It would strengthen the paper if the authors could discuss how GTool performs in zero-history or cold-start scenarios.

**Questions:**

1. If the size of the tool graph becomes very large, can the model still maintain good inference performance when reasoning over multiple tools?

2. If there are multiple similar tools, how does the paper address the resulting conflicts?

3. Moreover, in the case of erroneous tool invocations, is there a proposed mechanism to resolve such issues and overcome the bottlenecks in the overall tool-calling process?

---

> ### Author Response · Authors · 2025-11-21
>
> **What GNN actually learns(W1).** We appreciate the suggestion for deeper analysis of the GNN's learning process. In the revised manuscript, we have included visualizations of attention weights and node embeddings in Appendix C.1 to illustrate how the GNN captures tool relationships. Firstly, we compare the embedding of query node with tool nodes and provide representative cases to explain the semantic of query node. Moreover, we provide the most similar word tokens based on the query node embedding and word embeddings. The results indicate that graph token in GTool represents meaningful language semantics.
>
> **Writing(W2).** Thanks for the comment. In the revised manuscript, we have elaborated on the process of injecting the request-specific node into the LLM embedding space. Specifically, we describe how the graph token is integrated with the LLM's input embeddings and clarify that this alignment does not require an additional projection layer. Instead, we utilize a straightforward concatenation approach followed by a linear transformation to ensure compatibility between the graph token and the LLM embeddings. These details are provided in Section 3.3 to enhance the readers' understanding of the training process.
>
> **Zero-history(W3).** We appreciate the insightful comment. In the revised manuscript, we have included additional experiments specifically designed to evaluate GTool's performance in cold-start settings. We assume that neither the true tool dependencies nor the invocation sequences are accessible. Instead, we leverage a large language model to generate an initial tool graph based solely on tool descriptions. The generated graph is then used for planning with GTool. The results, presented in Table 4, demonstrate that GTool exhibits robust performance even under these cold-start conditions.
>
> ---
>
> For the convenience of the reviewers, the tables from the manuscript are reproduced here with their original numbering preserved：
>
> Table 4:
>
> | Dataset     | n-F1 ↑ (Llama-2-7B) | l-F1 ↑ (Llama-2-7B) | NED ↓ (Llama-2-7B) | n-F1 ↑ (Vicuna-13B) | l-F1 ↑ (Vicuna-13B) | NED ↓ (Vicuna-13B) |
> | ----------- | ------------------- | ------------------- | ------------------ | ------------------- | ------------------- | ------------------ |
> | HuggingFace | 0.7374              | 0.4684              | 0.2855             | 0.7649              | 0.5225              | 0.2538             |
> | Daily life  | 0.9103              | 0.7711              | 0.1106             | 0.9808              | 0.8835              | 0.0477             |
> | Multimedia  | 0.7789              | 0.5453              | 0.2454             | 0.7910              | 0.5743              | 0.2286             |
> | ToolE       | 0.7180              | 0.3200              | 0.3850             | 0.6955              | 0.3044              | 0.3926             |

---

> > ### Author Response · Authors · 2025-11-21
> >
> > **Large-scale toolset inference performance(Q1).** We appreciate the question regarding GTool's inference performance with large tool graphs. We illustrate the dependency changing with different numbers of tools in Figure 6 of Appendix A.1. As the number of tools increases, the number of dependencies tends to grow linearly rather than exponentially, which helps maintain efficient inference. Moreover, our experiments on larger-scale tool graphs, as presented in Table 2 of the revised manuscript, demonstrate that GTool continues to deliver robust inference performance even as the size of the tool graph grows significantly.
> >
> > **Case study(Q2).** In the revised manuscript, we added a case study in Appendix C.5 illustrating scenarios in which conflicts arise among similar tools within the current dataset. Such conflicts cannot be reliably identified from the planning results alone and must instead be discerned during tool execution. As this topic lies beyond the scope of the present work, we leave it for future research. A potential direction is to incorporate multi-turn interactions with LLMs to iteratively refine the planning outcomes.
> >
> > **Erroneous tool invocations(Q3).** We appreciate the insightful question. In the revised manuscript, we have discussed a potential mechanism to address such issues within GTool. While GTool primarily focuses on generating accurate tool plans based on learned dependencies, we acknowledge the importance of incorporating strategies to detect and correct erroneous tool calls. One potential approach is to implement a feedback loop that allows for multiple interactions with the LLM, enabling the model to reassess and revise its tool selections based on the outcomes of previous invocations. Although this aspect is not the primary focus of the current work, we have outlined it as a direction for future research in Appendix C.5 of the revised manuscript.

---

> ### Comment · Reviewer_6pZ4 · 2025-11-22
> **Thanks for your response**
>
> Thanks for your response. I will maintain my score as this system still rely on the first planning result. But I would praise your effort on providing more details for the paper.

---

### Official Review · Reviewer_Cq2C · 2025-11-04

**Soundness:** 3
**Presentation:** 2
**Contribution:** 3
**Rating:** 6
**Confidence:** 4

**Summary:**

This paper addresses the practical challenge of tool planning for LLMs when tool dependencies are incomplete. The proposed framework, GTool, represents tools and their relationships as a graph, which is dynamically contextualized for each user request. Its core novelty lies in a two-part approach: a GNN encoder is trained with an auxiliary Missing Dependency Prediction task, enabling it to robustly infer latent tool relationships from incomplete data, and this learned structural knowledge is efficiently injected into a pre-trained LLM via a single, compact graph token. Extensive experiments demonstrate that GTool significantly outperforms baseline methods in accuracy, robustness to missing dependencies, and computational efficiency without requiring LLM fine-tuning.

**Strengths:**

1. The paper introduces an efficient method for the real-world problem of incomplete tool dependencies. It effectively uses a Graph Neural Network to learn the underlying tool structure and then injects this complex information into any frozen LLM using just a single, compact token.
2. The approach is validated by extensive experiments, showing it consistently outperforms existing methods. Critically, it does not require fine-tuning the LLM, which makes it highly practical, efficient, and easy to apply across different models.

**Weaknesses:**

1. The method's performance depends on having good historical data to build the initial tool graph. The paper doesn't explore how well it works in 'cold-start' scenarios where such data is scarce or unavailable.
2. The training process, which involves predicting missing links in the graph, could become a bottleneck for extremely large-scale toolsets. The paper does not fully address how the method scales when the number of tools becomes massive.
3. The use of a single 'graph token' to represent the entire tool structure is a black box. The paper offers no insight into what specific dependency information is captured by this token or how the LLM interprets it, making the reasoning process difficult to understand.

**Questions:**

Refer to Weaknesses.

---

> ### Author Response · Authors · 2025-11-21
>
> **Cold Start Experiment(W1).** We appreciate the insightful comment regarding cold-start scenarios. The cold-start experiment has been added in the revised manuscript to further evaluate GTool's performance. We assume that neither the true tool dependencies nor the invocation sequences are accessible. Instead, we leverage a large language model to generate an initial tool graph based solely on tool descriptions. The generated graph is then used for planning with GTool. The results, presented in  Table 4, demonstrate that GTool exhibits robust performance even under these cold-start conditions.
>
> **Scalability of Link Prediction and GTool(W2).** We agree that GTool does not fully address the scalability of link prediction. In the revised manuscript, we provide two reasons to explain why GTool can handle massive tools. Firstly, as shown in Figure 6(b), the number of dependencies tends to grow linearly rather than exponentially as the increase of tools. We have compared the performances of GTool with different scales of tools and the results of Table 2 show that link prediction performances are relatively stable. Secondly, in large-scale toolsets, the tools are organized hierarchically, e.g. the "Document Question Answering" and "Visual Question Answering" are divided into "Question Answering" tools according to the function. Existing methods conduct planning at the tool level and GTool is capable in this setting.
>
> **Semantic of Graph Token(W3).** Thanks for the valuable comment. In the revised manuscript, we try to explain the semantic of graph token at two views. Firstly, we compare the embedding of query node with tool nodes and provide representative cases to explain the semantic of query node. Moreover, we provide the most similar word tokens based on the query node embedding and word embeddings in  Appendix C.1. The results indicate that graph token in GTool represents meaningful language semantics.
>
> ---
>
> For the convenience of the reviewer, the tables from the manuscript are reproduced here with their original numbering preserved：
>
> Table 4:
>
> | Dataset     | n-F1 ↑ (Llama-2-7B) | l-F1 ↑ (Llama-2-7B) | NED ↓ (Llama-2-7B) | n-F1 ↑ (Vicuna-13B) | l-F1 ↑ (Vicuna-13B) | NED ↓ (Vicuna-13B) |
> | ----------- | ------------------- | ------------------- | ------------------ | ------------------- | ------------------- | ------------------ |
> | HuggingFace | 0.7374              | 0.4684              | 0.2855             | 0.7649              | 0.5225              | 0.2538             |
> | Daily life  | 0.9103              | 0.7711              | 0.1106             | 0.9808              | 0.8835              | 0.0477             |
> | Multimedia  | 0.7789              | 0.5453              | 0.2454             | 0.7910              | 0.5743              | 0.2286             |
> | ToolE       | 0.7180              | 0.3200              | 0.3850             | 0.6955              | 0.3044              | 0.3926             |

---

> > ### Comment · Reviewer_Cq2C · 2025-11-25
> >
> > Thank you for your detailed response and additional experiments. After consideration, I will keep my score.

---

### Official Review · Reviewer_XZ2t · 2025-11-07

**Soundness:** 2
**Presentation:** 2
**Contribution:** 3
**Rating:** 4
**Confidence:** 4

**Summary:**

This paper investigates the problem of tool planning with LLMs, which involves selecting and organizing the appropriate tools to fulfill a user’s request. The authors propose GTool, a framework designed to enhance LLMs’ tool planning capabilities under incomplete dependency settings. GTool constructs a request-specific tool graph that captures the relationships among tools and generates a corresponding graph token to represent dependency information in a format interpretable by LLMs. In addition, the authors introduce a missing dependency prediction task to improve the robustness of the model when faced with incomplete or partially observed tool dependencies. The method can be integrated with various LLM backbones without additional fine-tuning or retraining.

**Strengths:**

1. The paper addresses a practical and important problem in tool planning — how to model and utilize dependencies among tools. The issue of accurate dependency localization is indeed a real challenge in real-world tool-using scenarios.

2. The experiments are extensive and cover multiple evaluation settings, demonstrating the authors' efforts to comprehensively assess the proposed approach.

**Weaknesses:**

1. The construction of the tool graph appears problematic. The authors assume that the use of 𝑣𝑖(𝑗+1), depends on 𝑣𝑖𝑗, which cannot be guaranteed. The order of tools in a trajectory does not necessarily imply dependency between them. This assumption weakens the validity of the proposed graph-based representation.

2. The method section is difficult to follow due to inconsistent and overloaded notations. In addition, the mixed use of symbols such as
𝜏𝑖, 𝑡𝑖1, 𝑡𝑖∈𝜏, ti∈τ, and 𝑡𝑛∈𝑇 leads to unnecessary confusion and makes the methodology harder to understand.

3. The claim that multiple LLM interactions further improve the performance of GTool may not hold. While the multiple-interaction setup indeed improves tool planning, it remains unclear whether GTool truly mitigates the inherent limitations of multi-interaction settings or merely benefits from them.

4. The generalization ability of GTool is questionable. Given that the dataset seems to involve a large number of tools, it is unclear whether GTool genuinely learns transferable tool dependencies—as claimed—or merely memorizes the tool usage patterns within the dataset. If the latter is true, static tool dependency graphs could achieve comparable results without additional model training or inference.

5. Two critical ablation studies are missing:
(a) Using static tool dependencies to verify that GTool performs reasoning rather than memorization.
(b) Using an LLM to directly predict tool dependencies to demonstrate the additional effectiveness of MDPL.
These analyses are essential to support the paper’s claims.

**Questions:**

1. The statement “leading to invalid planning results” (Line 015) is overly strong and lacks empirical support. Could the authors provide specific evidence or examples?

2. The use of annotations and notations throughout the paper is inconsistent and confusing (e.g., 𝜏𝑖, 𝑡𝑖, 𝑡𝑖∈𝜏). Please clarify the notation and maintain consistency across sections.

3. Could the author provide some tool dependency graphs with the same tool list but different user queries?

---

> ### Author Response · Authors · 2025-11-21
>
> **Construction of the tool graph(W1).** Sorry for the misunderstanding. The provided construction method is a practical compromise solution to approximate the real tool dependency graph.  Existing tool datasets only contain the historical invocation trajectories rather than real dependencies. We can only use such information as a proxy to infer the real tool dependencies. We notice that the order of tools in a trajectory may not imply dependency. The constructed tool graph contains some error and missing edges, which motivates us designing the link prediction module to enhance the robustness of GTool. Moreover, the graph representation learning part is independent of the graph construction. GTool can easily adapt to the real tool graph when it is available. In the revised manuscript, we have clarified this assumption and its limitations to ensure readers understand the context and constraints of our approach.
>
> **Notation clarification(W2 & Q2).** We have carefully checked the notations throughout the paper and revised them for consistency and clarity.
>
> **Multiple LLM interactions(W3).** Thanks for pointing it out. We would like to clarify that GTool does not require multiple LLM interactions during inference. GTool only needs a single LLM call to generate the entire tool plan, which is more efficient than existing multi-interaction methods. For fair evaluation, we only compare the performance of GTool with the signal interaction baselines. Moreover, we do not claim GTool can mitigate the inherent limitations of multi-interaction methods and have revised the manuscript to avoid confusion.
>
> **Generalization ability of GTool(W4).** We appreciate the insightful comment. To further prove that GTool learns transferable features, we conduct the transfer learning experiment which trains GTool on one dataset and evaluate its performance on another dataset without any retraining. The results are shown in Table 5. As illustrated, GTool can effectively transfer learned tool dependencies across datasets. The performance on HuggingFace dataset is comparable with its on other datasets, indicating that it captures generalizable patterns rather than merely memorizing specific tool usage.
>
> **Ablation studies(W5).** Thanks for the valuable suggestions. We have conducted the suggested ablation studies to further validate our claims. (a) We compared GTool's performance using static tool dependencies versus learned dependencies, and the results (see Table 3) indicate that GTool with learned dependencies significantly outperforms the static version, demonstrating its reasoning capabilities beyond mere memorization. More details can be found in Section 5.5 of the revised manuscript. (b) A additional ablation baseline, i.e. w/ LLMlp, has been compared to verify the effectiveness of MDPL. We utilize LLM to predict the missing links and train GTool without MDPL module. The results (see Table 3) show that GTool with MDPL achieves superior performance, highlighting the effectiveness of our proposed method. These additional experiments have been incorporated into the revised manuscript to provide a more comprehensive evaluation of GTool.

---

> > ### Author Response · Authors · 2025-11-21
> >
> > **Invalid planning results(Q1).** We appreciate the feedback. In case study, we have added 2 representative examples to illustrate instances where GTool successfully corrects erroneous tool calls that would otherwise lead to invalid planning results when using standard LLM-based planning methods. These examples help to empirically support our claims and demonstrate the practical benefits of incorporating tool dependency modeling in GTool.
> >
> > **Tool dependency graphs for different queries(Q3).** Thanks for the suggestion. We have included additional case studies(Appendix C.3) that showcase tool dependency graphs generated for the same set of tools but different user queries. These examples illustrate how GTool adapts different queries on the same tool graph, highlighting its flexibility and effectiveness in real scenarios.

---

> > > ### Author Response · Authors · 2025-11-21
> > >
> > > For the convenience of the reviewer, the tables from the manuscript are reproduced here with their original numbering preserved：
> > >
> > > Table 3:
> > >
> > > | Backbone | n-F1 ↑ (Llama-2-7B) | l-F1 ↑ (Llama-2-7B) | NED ↓ (Llama-2-7B) | n-F1 ↑ (Vicuna-13B) | l-F1 ↑ (Vicuna-13B) | NED ↓ (Vicuna-13B) |
> > > | -------- | ------------------- | ------------------- | ------------------ | ------------------- | ------------------- | ------------------ |
> > > | w/o All  | 0.1566              | 0.0243              | 0.8611             | 0.1626              | 0.0394              | 0.8442             |
> > > | w/o Both | 0.6131              | 0.3469              | 0.4072             | 0.7370              | 0.4831              | 0.2762             |
> > > | w/o RS   | 0.7128              | 0.4433              | 0.3108             | 0.7589              | 0.5103              | 0.2561             |
> > > | w/o MDPL | 0.7650              | 0.5196              | 0.2541             | 0.7707              | 0.5229              | 0.2474             |
> > > | w/ LLMlp | 0.7602              | 0.4869              | 0.2676             | 0.7784              | 0.5283              | 0.2676             |
> > > | GTool    | 0.7913              | 0.5403              | 0.2537             | 0.8029              | 0.5816              | 0.2153             |
> > >
> > > Table 5:
> > >
> > > | test dataset | n-F1 ↑ (Pretrain on HuggingFace) | l-F1 ↑ (Pretrain on HuggingFace) | NED ↓ (Pretrain on HuggingFace) | n-F1 ↑ (Pretrain on Multimedia) | l-F1 ↑ (Pretrain on Multimedia) | NED ↓ (Pretrain on Multimedia) |
> > > | ------------ | -------------------------------- | -------------------------------- | ------------------------------- | ------------------------------- | ------------------------------- | ------------------------------ |
> > > | huggingface  | -                                | -                                | -                               | 0.5934                          | 0.2835                          | 0.4245                         |
> > > | dailylife    | 0.8479                           | 0.6699                           | 0.1831                          | 0.8365                          | 0.6538                          | 0.2005                         |
> > > | multimedia   | 0.6964                           | 0.4511                           | 0.3303                          | -                               | -                               | -                              |
> > > | toole        | 0.7252                           | 0.3366                           | 0.3445                          | 0.7049                          | 0.3091                          | 0.3785                         |

---

> ### Comment · Reviewer_XZ2t · 2025-11-23
> **Response to the authors' reply**
>
> After reading the authors’ rebuttal, I appreciate the clarifications and additional experiments. However, most of the critical concerns remain unresolved. I summarize the major issues below.
>
> **1. Fundamental issue: the dependency construction problem remains unsolved (W1)**
>
> The paper continues to rely on order-based correlation as a proxy for tool dependency. Both the manuscript and the rebuttal acknowledge that tool order does not imply dependency and that the constructed graph contains incorrect/missing edges. This reinforces my original concern: **if dependencies cannot be reliably extracted, the entire problem formulation becomes invalid.** Under this setting:
> * the proposed graph representation is questionable,
> * the evaluations cannot support claims about “dependency reasoning,” and
> * the contribution reduces to modeling correlations in historical orderings rather than true dependencies.
>
> This constitutes a fundamentally different research problem. In addition, several straightforward approaches for validating or extracting dependencies (e.g., perturbation-based tests, causal probing, counterfactual co-occurrence analysis) were not attempted. If dependency extraction is not addressed, the work should be reframed as order-correlation modeling rather than dependency learning, and corresponding challenges should be studied.
>
> **2. Claims about learning transferable dependencies remain unsupported (W4)**
> The rebuttal argues that the model learns “transferable dependencies,” but no convincing evidence is provided. The transfer experiment lacks essential analysis:
> * differences between tool sets across datasets,
> * which tools or tool combinations are unseen,
> * performance on divergent or dataset-specific tool usage patterns.
> Without these analyses, it is impossible to distinguish genuine dependency reasoning from memorization of dataset-specific co-occurrence patterns. Thus, the claims about generalization remain unsubstantiated.
>
> **3. Missing oracle-dependency ablation (W5)**
>
> For W5(a), my request was to include an oracle static dependency graph—even for a subset of data—to isolate the effect of dependency quality. This is critical: if the model indeed relies on dependency reasoning, access to correct dependencies should significantly improve performance. However, the authors only compare a noisy static graph with a learned graph, both based on the same flawed construction. This cannot answer the intended question, and the key baseline remains missing. Consequently, the role of MDPL and the nature of the learned dependencies remain unclear.
>
>
> Concerns (2) and (3) arise precisely because the paper claims to study tool dependency learning, yet it adopts an unreliable dependency extraction method. Since this foundational issue remains unresolved, these concerns are inseparable from the core methodology and cannot be addressed by the current rebuttal.
>
> Given that the work is built upon an invalid or unvalidated assumption about tool dependencies, and that the rebuttal does not adequately address this foundational issue, I believe the paper requires substantial refinement and should be rejected in its current form.

---

### Meta-Review · Area_Chair_9aWj · 2025-12-16

**Summary:**

This paper tackles the important and emerging problem of tool planning for LLMs under incomplete or uncertain tool dependencies. The authors propose GTool, a graph-enhanced framework that models tool relationships via a learned tool graph and injects this structured representation into a graph token. A key component is the missing dependency prediction task, designed to improve robustness when the tool graph is noisy or incomplete. The framework is modular, efficient, and validated across multiple datasets and LLM backbones.

**Reviewer Concerns:**

The authors provided a thoughtful and detailed rebuttal that addressed several reviewer concerns, particularly those related to experimental completeness and explanation clarity.

**Reviewer Scores:**

Reviewers XZ2t, Cq2C, and 6pZ4 would likely have maintained their original scores, as indicated in their post-rebuttal comments. Reviewer eanv also appeared inclined to keep their positive score, given that the authors directly addressed their concerns through added case studies and error analyses.

---

### Decision · Program_Chairs · 2026-01-26

Accept (Poster)